# Structural basis of the effect of activating mutations on the EGF receptor

Ioannis Galdadas[1,2], Luca Carlino[3], Richard A Ward[3], Samantha J Hughes[3], Shozeb Haider[4], Francesco Luigi Gervasio[1,2,5,6]*

[1]Department of Chemistry, University College London, London, United Kingdom; [2]Institute of Pharmaceutical Sciences of Western Switzerland, University of Geneva, Geneva, Switzerland; [3]Oncology R&D, AstraZeneca, Cambridge, United Kingdom; [4]UCL School of Pharmacy, University College London, London, United Kingdom; [5]Institute of Structural and Molecular Biology, University College London, London, United Kingdom; [6]Pharmaceutical Sciences, University of Geneva, Geneva, Switzerland

**Abstract** Mutations within the kinase domain of the epidermal growth factor receptor (EGFR) are common oncogenic driver events in non-small cell lung cancer. Although the activation of EGFR in normal cells is primarily driven by growth-factor-binding-induced dimerization, mutations on different exons of the kinase domain of the receptor have been found to affect the equilibrium between its active and inactive conformations giving rise to growth-factor-independent kinase activation. Using molecular dynamics simulations combined with enhanced sampling techniques, we compare here the conformational landscape of the monomers and homodimers of the wild-type and mutated forms of EGFR ΔELREA and L858R, as well as of two exon 20 insertions, D770-N771insNPG, and A763-Y764insFQEA. The differences in the conformational energy landscapes are consistent with multiple mechanisms of action including the regulation of the hinge motion, the stabilization of the dimeric interface, and local unfolding transitions. Overall, a combination of different effects is caused by the mutations and leads to the observed aberrant signaling.

**\*For correspondence:**
francesco.gervasio@unige.ch

## Introduction

The epidermal growth factor receptor (EGFR) is a cell surface receptor, which regulates cell proliferation and differentiation by phosphorylating downstream signaling proteins (*Sigismund et al., 2018*; *Wee and Wang, 2017*). Its physiological activity is tightly regulated and genetic mutations that lead to its uncontrolled activation have been associated with cancer development (*Sharma et al., 2007*; *Sigismund et al., 2018*). Indeed, acquired somatic DNA alterations on the gene which encodes EGFR are found in 14–32% of non-small cell lung cancer (NSCLC) cases (*Collisson et al., 2014*; *Zhang et al., 2016*) and are correlated with adverse prognosis (*Hirsch et al., 2003*; *Sharma et al., 2007*). EGFR in-frame deletions in exon 19 are the most prevalent of EGFR kinase domain mutations accounting for 45% of EGFR mutations in NSCLC, followed by the L858R missense mutation in exon 21, which accounts for approximately 40–45% of EGFR kinase domain mutations (*Sharma et al., 2007*). Exon 20 insertions (Ex20Ins) collectively account for the third most common category of EGFR mutations found in NSCLC and are detected in 4–10% of the cases (*Arcila et al., 2013*; *Yasuda et al., 2012*).

Consistent with their purported role in the etiology of NSCLC, recent studies have shown that β3-αC ΔELREA, L858R, A763-Y764insFQEA, and D770-N771insNPG mutated EGFR proteins are oncogenic in both cell cultures and transgenic mouse models (*Ji et al., 2006*; *Politi et al., 2006*; *Sordella et al., 2004*; *Xu et al., 2007*; *Yasuda et al., 2013*). These mutations increase the kinase

activity of EGFR and eliminate the dependence of the cancer mutants on EGF binding for activation, leading to stimulation of downstream pro-survival and cell-growth pathways, and consequently conferring oncogenic properties on EGFR. It should be noted though that the different mutations lead to different levels of activation, with ΔELREA-EGFR exhibiting the highest catalytic activity among them (29-fold higher than that for wild-type EGFR) (*Gilmer et al., 2008*), followed by the L858R (23-fold) (*Gilmer et al., 2008*), the A763-Y764insFQEA (9-fold) (*Yasuda et al., 2013*), and lastly the D770-N771insNPG (5-fold) (*Yasuda et al., 2013*).

Like other eukaryotic protein kinase domains, the kinase domain of EGFR consists of a smaller N-terminal lobe (N-lobe) and a larger C-terminal lobe (C-lobe), with the ATP-binding site located between them. The N-lobe is composed of five β-strands and a helix referred to as αC-helix, whereas the C-lobe is predominantly helical. A glycine-rich loop located on the N-lobe, also known as the 'P-loop', extends over the top of the ATP-binding site and along with a number of conserved residues facilitate the coordination of Mg-ATP in the catalytic pocket. Specifically, a conserved lysine (K745) binds to the α- and β-phosphates of ATP, underlying its importance for efficient phosphate transfer (*Honegger et al., 1987*; *Robinson et al., 1996*). This lysine is buried deep within the interlobal cleft, where it is stabilized and oriented properly for catalysis by a conserved glutamate (E762) on the αC-helix. When these two amino acids are mutated, the kinase is unable to bind ATP leading to reduced activity (*Honegger et al., 1987*; *Jura et al., 2009*). The aspartate (D855) of a conserved Asp–Phe–Gly (DFG) motif found at the N-terminal end of the activation loop (A-loop), is also a key regulatory element as it participates in the binding of the catalytic ion. The direct contact of the αC-helix with the N-terminal region of the activation loop together with the lysine-glutamate ion pair couple the conformations of the αC-helix and A-loop to ATP binding.

The full length of the A-loop is often unresolved in crystal structures and not visible in NMR spectra, which is indicative of its high flexibility and conformational variability. Broadly speaking, four conformations of the activation loop are seen in available kinase structures: active, Src-like inactive, substrate competitive, and detached (*Figure 1—figure supplement 1*). The active conformation of the activation loop, which includes the formation of a β-strand (β9-strand) with residues of the C-lobe, is shared by almost all protein kinases, whereas its inactive conformations vary among different kinases. The Src-like inactive conformation, named after the resemblance of the conformation to the inactive conformations of the Src kinase, is common in many kinases, including Src, Abl, CDK2, and EGFR and involves the formation of a two-turn helix at the N-terminal part of the activation loop. The substrate-competitive conformation, which features an activation loop positioned toward the hinge which shields the substrate-binding site, is adopted by IRK, Abl, Src, and p38 MAP kinases, among others. In the detached conformation, the activation loop assumes an extended conformation and the short αEF-helix is detached from the main body of the kinase (*Shan et al., 2013*). This conformation is typically seen in kinase dimers in which the activation loop engages in intermolecular interactions, which may underlie trans-autophosphorylation (*Pike et al., 2008*).

It has been established that the activation of EGFR is associated with several rearrangements of different structural elements, apart from the ones seen in the activation loop, including that of the αC-helix of the N-lobe of the kinase domain (*Endres et al., 2011*; *Kovacs et al., 2015*). In the active conformation, the αC-helix adopts the so-called αC-in conformation, which is characterized by the full formation of the so-called catalytic spine and the formation of the functionally important KE salt bridge between K745 and E762. The helix also adopts the αC-out conformation in inactive forms of EGFR, in which the αC-helix is displaced, and the KE salt bridge is broken.

Long-timescale molecular dynamics simulations supported by H/D exchange experiments have identified the presence of a functionally important conformation of EGFR, in which the αC-helix exhibits local disorder (partial unfolding) (*Shan et al., 2012*). This intrinsically disordered nature of the αC-helix has been postulated to be essential for the regulation of lateral propagation of EGFR signal in the absence of external stimulation through the formation of asymmetric homodimers or higher-order oligomers, which is an essential step for the activation of EGFR (*Zanetti-Domingues et al., 2018*). Specifically, phosphorylation of Y869 has been shown to stabilize the αC-helix, and therefore the region that the receiver EGFR monomer binds to, lowering thus the energy threshold for EGFR multimerization and leading to propagation of the signal (*Shan et al., 2012*).

The active to inactive transitions of the above described structural elements are fundamental in the regulation of EGFR, and different mutations have been associated with dysregulation of these conformational changes in cancer. To better understand the effect of the most frequently observed

and clinically relevant ∆ELREA and L858R mutations (*Figure 1*), we performed here long-timescale atomistic molecular dynamics (MD) simulations and enhanced sampling free energy calculations with a recently developed force field. The chosen force field has been shown to be able to reproduce reliably the dynamics of both well-folded and partially unfolded domains (*Robustelli et al., 2018*). As most Ex20Ins mutations have been shown to render EGFR resistant to most tyrosine kinase inhibitors (TKIs) (*Yasuda et al., 2012*; *Ward et al., 2021*), with the exception of the relatively rare A763-Y764insFQEA insertion, we tried to rationalize the behavior of a prototypical Ex20Ins, D770-N771insNPG, and of A763-Y764insFQEA, for which structural and biochemical data is available. The unbiased MD simulations of the isolated monomers and the asymmetric homodimers collectively lasted 36 µs for the WT and the mutant forms (see Materials and methods, *Table 1*). As these

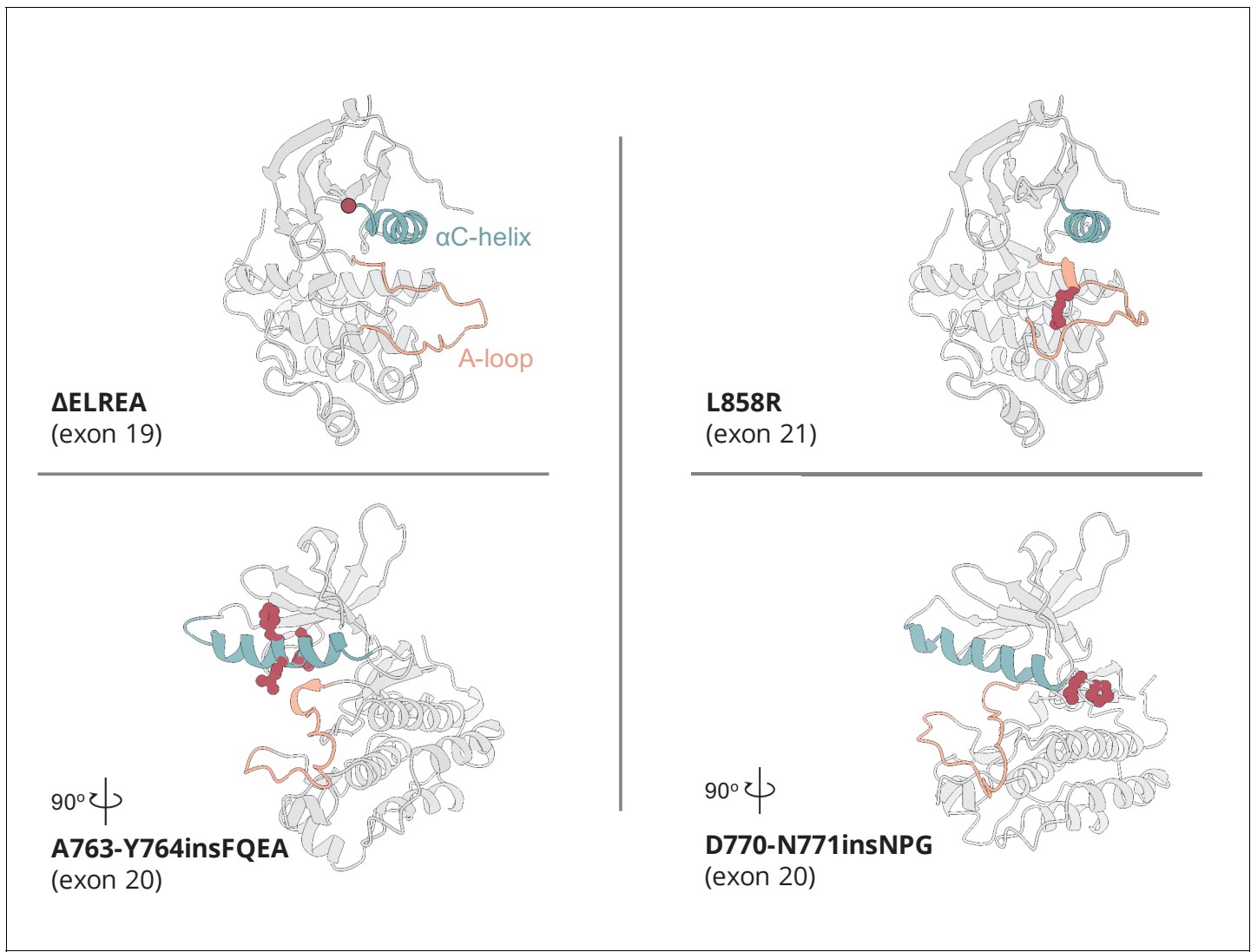

**Figure 1.** Active conformation of the somatic mutations on EGFR kinase domain studied here. The αC-helix is depicted in cyan, while the A-loop in orange. The point of deleted residues in the ∆ELREA mutant is indicated with a red dot. In the case of the point mutation L858R, and the two Ex20Ins, the residues associated with the mutation are depicted in red spheres.

The online version of this article includes the following figure supplement(s) for figure 1:

**Figure supplement 1.** Conformational variability of the αC-helix and A-loop of the kinase domain.

**Figure supplement 2.** Simple moving averaged time series of the distance between two salt-bridge-forming residue pairs K745-E762 and D855-E762 over the course of the unbiased simulations starting from the monomeric active or Src-like inactive conformation.

**Figure supplement 3.** Cluster of hydrophobic residues that stabilize the two-turn helix of the A-loop and the Src-like inactive conformation overall.

**Figure supplement 4.** Position of the αC-helix in each monomer of the simulated dimers.

**Table 1.** Summary of the unbiased simulations of the monomeric EGFR.
The reported times correspond to the simulation time of each independent unbiased simulation.

| System | Starting conformation | Total simulation time (ns) |
|---|---|---|
| WT | active | 1000 |
| | active | 1000 |
| | Src-like inactive | 1000 |
| | Src-like inactive | 1000 |
| L858R | active | 1000 |
| | Src-like inactive | 1000 |
| A763-Y764insFQEA | active | 1000 |
| | Src-like inactive | 1000 |
| D770-N771insNPG | active | 1000 |
| | Src-like inactive | 1000 |
| ΔELREA | active | 1000 |
| | Src-like inactive | 1000 |
| [1ex] | | |

simulations hinted at slow motions that could not be sampled even by long MD simulations, we also applied enhanced sampling techniques, and in particular, parallel tempering metadynamics (PTmetaD) simulations. PTmetaD simulations allow the exploration of biologically meaningful conformational changes of kinases that take place on time scales longer than those accessible through standard MD simulations and reconstruct the associated free energy landscapes (*Sutto and Gervasio, 2013*; *Marino et al., 2015*; *Kuzmanic et al., 2017*).

## Results

### WT *apo* populates inactive states with a semi-closed A-loop and a partially disordered αC-helix

In the simulations that we initiated from the active conformation of the *apo*, unphosphorylated EGFR monomer, the wild-type (WT) EGFR repeatedly departed from its initial αC-in conformation and transitioned to an αC-out-like conformation within ~100 ns (*Figure 1—figure supplement 2*) that resembles the conformation seen in the DFG-out inactive state (PDB ID: 2RF9). However, although the K745-E762 salt bridge was broken, the repositioning of the αC-helix was not sufficient for the A-loop to adopt the characteristic two-turn helix that is found in the Src-like inactive conformation. In fact, the A-loop remained extended throughout the simulations, which suggests the presence of an energy barrier that needs to be crossed for the A-loop to break the β9-strand and form the characteristic two-turn helix (*Sutto and Gervasio, 2013*). This behavior is in line with the known activation mechanism of EGFR according to which dimerization is important for keeping the αC-helix in the αC-in conformation (*Shan et al., 2012*) and suggests that the presence of ATP and phosphorylation may also be critical for full stabilization of the active conformation.

Indeed, our unbiased simulations of the asymmetric WT-EGFR dimer further support the notion that the active conformation is not adopted spontaneously in solution but is promoted by the interactions between the kinase domains in the asymmetric configuration. In particular, we see that the αC-helix is kept in the αC-in conformation predominately on the receiver monomer (*Figure 1—figure supplement 4*). On the contrary, the αC-helix of the activator monomer is flexible enough to adopt αC-out conformations, similar to the ones the αC-helix adopts in the monomeric form of EGFR. Unlike the behavior of the αC-helix that we see in the asymmetric dimer, in the simulations of the symmetric dimer, in which both monomers are found in the Src-like inactive conformation, the αC-helix of both monomers remains in the αC-out conformations throughout the simulation (*Figure 1—figure supplement 4*).

The free energy landscape reconstructed from multiple replica metadynamics simulations shows three main minima. The shape and position of them are reassuringly equivalent to those we previously obtained with a different force field (*Sutto and Gervasio, 2013*). As expected, in the absence of a ligand or phosphorylation, these minima for the WT-EGFR correspond to inactive-like conformations (*Figure 2*). Specifically, the ensemble of conformations that corresponds to the deepest free energy minimum – (CV1, CV2) = (0.6, 0.16) – contains conformations in which the A-loop has adopted a semi-closed conformation that precludes the ATP-binding, while the αC-helix is mostly ordered and has adopted the αC-out conformation (*Figure 2*, basin α1). The conformations in this basin are consistent with crystal structures of the Src-like inactive state (e.g. PDB ID: 2GS7) not only

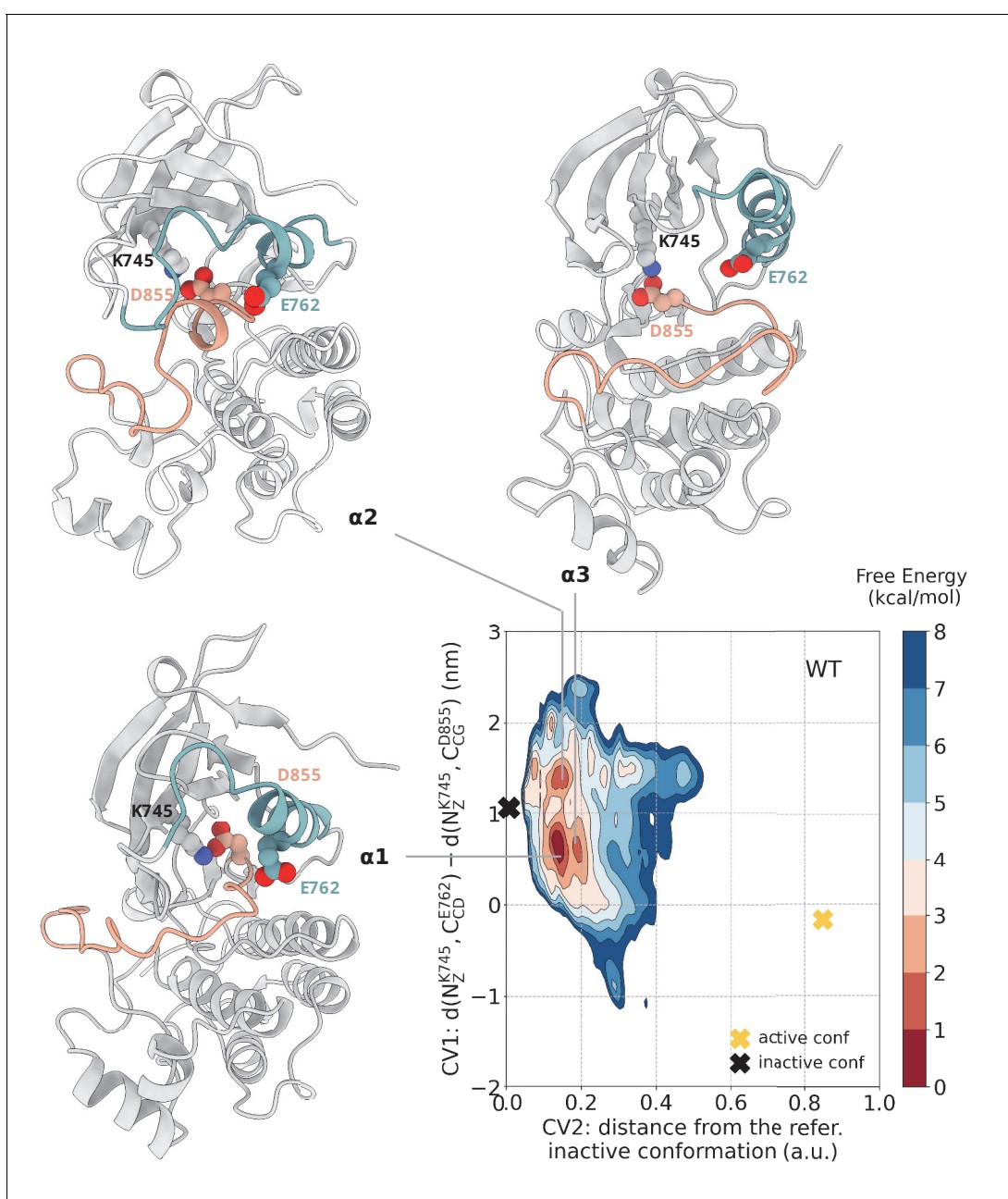

**Figure 2.** Free energy surface of the WT EGFR as a function of CV1 (distances between K745:E762 and K745:D855) and CV2 (distance from reference Src-like inactive structure). Representative structures for free energy minima are depicted in cartoon representation. A yellow cross indicates the position of the active conformation (PDB ID: 2GS2) in the explored CV space, while a black cross indicates the position of the Src-like inactive conformation (PDB ID: 2GS7) as reference.

in the backbone arrangement but also in the details of key interactions. In particular, in this ensemble of conformations, the salt bridge of E762 with K745 is broken, and a new one is formed between K745 and D855.

In one of the two secondary free energy minima - (CV1, CV2) = (1.4, 0.1) — the characteristic two-turn helix on the A-loop of the Src-like inactive conformation is formed and the αC-helix has rotated away from the C-lobe and adopted the αC-out conformation (*Figure 2*, basin α2). Notably, in the most populated state of this basin, the N-terminal region of the αC-helix is unfolded, in accordance with the predicted partially disordered nature of this region (*Shan et al., 2012*). In the other secondary free energy minimum – (CV1, CV2) = (0.6, 0.2) – the N-terminal end of the αC-helix is folded, and the helix has rotated out of the core of the protein, while the A-loop is still relatively unstructured (*Figure 2*, basin α3). Although the conformations in this basin resemble a lot the ones seen in the deepest energy minimum, they differ in the fact that the αC-helix points further away from the C-lobe, which results in a slight increase in the interlobal distance seen in the conformations of this basin. Interestingly, reprojection of the free energy with respect to the interlobal distance shows that the WT samples multiple conformations of different N/C separation (*Figure 7—figure supplement 4C*). Together, these two secondary minima may represent metastable states that are important during the transition from the active to the Src-like inactive conformation and reflect the necessary rearrangements for the deactivation to take place, that is unfolding of the N-terminal end of the αC-helix, increase in the interlobal distance and formation of the two turn helix on the A-loop. Despite the fact that the minimum energy path connecting the Src-like inactive to active conformation does not pass through the basin where the N-terminal end of the αC-helix is highly disordered (*Figure 7—figure supplement 3*), the 2 kcal/mol energy difference between the deepest minimum and this metastable state suggest that, under physiological conditions, this state should still be accessible.

## L858R stabilizes active-like conformations

It has been suggested that the mutation of L858 to arginine enhances the kinase activity by shifting the equilibrium toward the active conformation through disruption of the hydrophobic packing of L858 with L747, I759, and L861 (*Figure 1—figure supplement 3*) that is supposed to stabilize the Src-like inactive conformation (*Yasuda et al., 2013*). However, our unbiased simulation of the monomeric L858R starting from the Src-like inactive conformation showed that transient, favorable interactions of R858 with D855 and D837 of the DFG and HRD motifs, respectively, and F723 of the P-loop (*Figure 3—figure supplement 1*) stabilize the Src-like inactive conformation. On the contrary, in our unbiased simulations where L858R started from the active conformation, the side chain of R858 interacted mainly with E762 only (*Figure 3—figure supplement 1*).

The metadynamics simulations of the L858R mutant confirm the results that we obtained with an earlier force field (*Sutto and Gervasio, 2013*). The new free energy surface and the underlying structural ensemble is equivalent to the one obtained with the older force field, which bodes well for the robustness and accuracy of the results. The free energy landscape (*Figure 3*) shows that the deepest basin corresponds to an ensemble of conformations that are very close to the active conformation (*Figure 3*, basin δ1). The positively charged R858 was surrounded by a cluster of negatively charged residues (E758, E762, D837, and D855) and in this ensemble, the side chain of R858 fluctuated between two states in which it interacts with either D837 and D855, or E758 and E762. The interactions of R858 with D837 kept the A-loop in an extended, active-like conformation, in which the β9-strand on the A-loop is present. At the same time, as a result of the favorable interactions of R858 with E758 and E762, the αC-helix exhibited high secondary structure stability as opposed to the disorder seen in the αC-helix of the wild-type. Although the salt bridge of E762 with K745 is never present in this ensemble as is in the fully active conformation, the αC-helix is maintained close to the αC-in conformation, probably due to the interaction of R858 with E762. Overall, the presence of R858 favors the active conformation at the expense of the disordered state that is highly populated in the WT. Stabilization of the αC-helix through a mutation like L858R has been shown to promote EGFR dimerization and downstream signaling, which explains the activating nature of this mutation (*Shan et al., 2012*; *Zhang et al., 2006*). In a secondary minumum higher in energy (*Figure 3*, basin δ2), the side chain of R858 is found completely exposed to the solvent, but although the αC-helix is found in the αC-out conformation, the short helix on the A-loop is not formed. Instead, the A-loop assumes a semi-closed conformation.

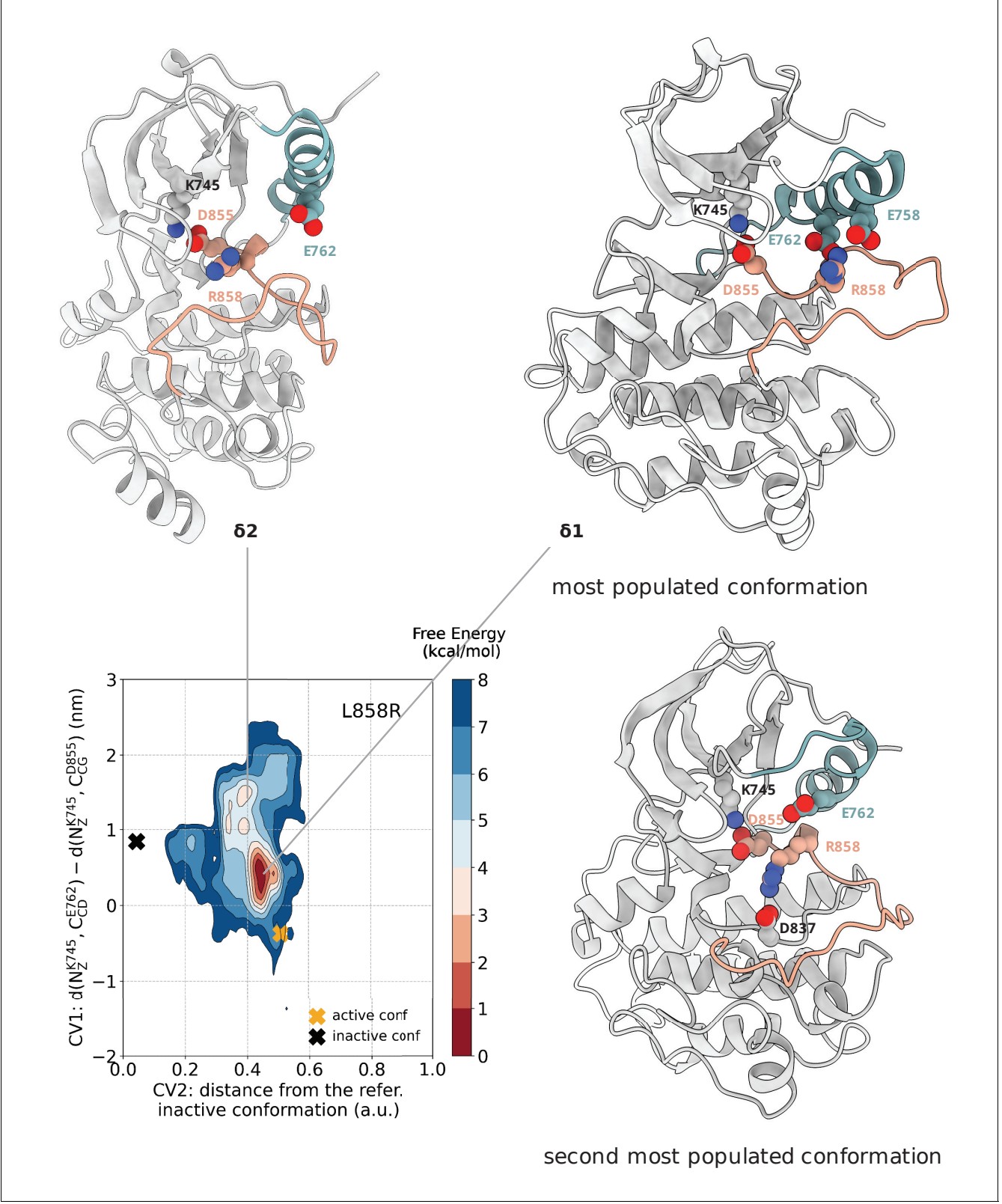

**Figure 3.** Free energy surface of the L858R mutant as a function of CV1 and CV2. Representative structures for two main conformations found in the deepest minimum are depicted in cartoon representation. The interaction of R858 with E758 and E762 is responsible for the secondary structure stability of the αC-helix, while the interaction of R858 with D837 and D855 prevents the formation of the two-turn helix on the A-loop. A yellow cross

*Figure 3 continued on next page*

*Figure 3 continued*

indicates the position of the active conformation (PDB ID: 2ITV) in the explored CV space, while a black cross indicates the position of the Src-like inactive conformation (homology model) as reference.

The online version of this article includes the following figure supplement(s) for figure 3:

**Figure supplement 1.** Important interactions of L858R with neighboring residues that are maintained over the course of the unbiased simulations of the monomeric L858R EGFR.

## D770-N771insNPG populates active-like and disordered αC-helix conformations

Based on the crystal structure of the D770-N771insNPG mutant in the active conformation, it has been postulated that the NPG insertion at the C-terminal end of the αC-helix, locks the helix in the αC-in conformation (Yasuda et al., ). Proline is a residue found commonly in turns and in the case of the D770-N771insNPG, the inserted P772 induces a turn that leads to the formation of a hydrogen bond between the backbone of D770 and the amide of the inserted G773 in the crystal structure. This backbone hydrogen bond orients, in turn, the side chain of D770 such that it forms a salt bridge with R779 (*Figure 4—figure supplement 1A*) that is maintained throughout the unbiased simulation initiated from the active conformation (*Figure 4—figure supplement 2A*), while the inserted N771 interacts with R837 of the E-helix. Interestingly, the same salt-bridge interaction is lost after about 300 ns in both replicas of the WT simulations (*Figure 4—figure supplement 2A*) and R776 forms a hydrogen bond with the backbone of A767 after (*Figure 4—figure supplement 2B*). Given the different behavior of the mutant and WT, one would expect these interactions to stabilize the αC-in conformation in the mutant, however, the K745-E762 salt bridge breaks almost immediately (*Figure 1—figure supplement 2*), suggesting that the αC-in conformation in which the mutant is found in the crystal structure is probably ligand-induced rather than mutation-induced.

The unbiased simulation of the D770-N771insNPG insertion that was initiated from the Src-like inactive conformation in the monomeric form suggests that as soon as the inserted triplet adopts an extended conformation where the backbone D770-G773 hydrogen bond is lost (*Figure 4—figure supplement 1B*), D770 starts interacting only sporadically with R779 (*Figure 4—figure supplement 2A*). In return, R779 forms a hydrogen bond with the carbonyl group of A767 (*Figure 4—figure supplement 1B*, *Figure 4—figure supplement 2B*), similar to the stable interaction seen in the simulations of the WT-EGFR in the Src-like inactive conformation (*Figure 4—figure supplement 2B*). Interestingly, R779H has been found to increase the phosphorylation of monomeric EGFR (*Ruan and Kannan, 2015*), underlying the role of R779 in the regulation of the αC-in and αC-out equilibrium. Our unbiased simulations suggest that the orientation that D770 adopts because of the inserted residues effectively prevents R779 from interacting with A767 that is seen in the αC-out conformation. This already shows the crucial role of the P771, G773, D770, R779, A767 network of interactions in the stabilization of different αC-helix conformations.

In the PTmetaD simulations and in the absence of a phosphate group on Y869 or a ligand, the αC-helix adopts the αC-out conformation where the K745-E762 salt bridge is broken, but the A-loop assumes an extended, active-like conformation. Mapping of the conformational space sampled during the unbiased monomeric simulations of the D770-N771insNPG on the FES shows that the deepest minimum corresponds to the CV space that the mutant samples in the simulations that we initiated from the active conformation (*Figure 7—figure supplement 2*). Projection of the FE on the (CV1, CV2) space results in a broad minimum around where the active conformation lies. Reprojection of the FE on the (CV1, CV3) space, however, separates the two dominant conformations that fall together in the same basin in the (CV1, CV2) projection of the FE (*Figure 4*).

In one of the two isoenergetic basins in the free energy surface (*Figure 4*, basin ε1), the side chain of the inserted N771 has flipped 180° with respect to the crystal structure and interacts with the side chain of R779. The mutation does not seem to be able to quench the intrinsic disorder of αC-helix fully, as can be seen from the unfolded N-terminal part of the helix in the representative conformation of the minimum (*Figure 4*). In the second basin (*Figure 4*, basin ε2), the D770-R779 salt bridge is formed, similar to the crystal structure, and so is the backbone hydrogen bond between D770 and G773, and the αC-helix is fully helical. However, unlike the crystal structure, the αC-helix is in the αC-out conformation.

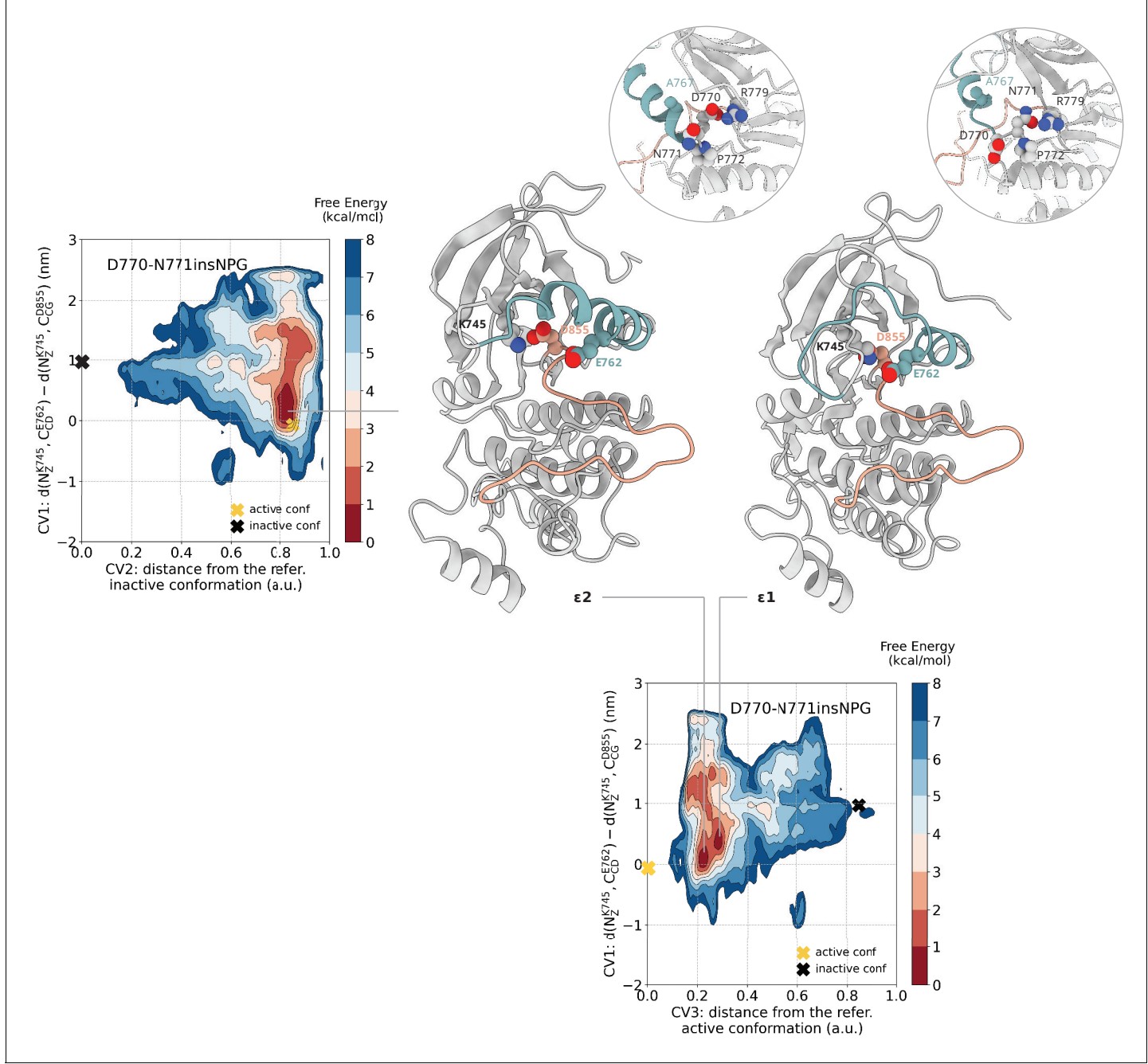

**Figure 4.** Free energy surface of the D770-N771insNPG mutant as a function of CV1 and CV2 as well as of CV1 and CV3. Representative structures of the two main conformations found in the deepest minima are depicted in cartoon representation. A yellow cross indicates the position of the active conformation (PDB ID: 4LRM) in the explored CV space, while a black cross indicates the position of the Src-like inactive conformation (homology model) as reference.

The online version of this article includes the following figure supplement(s) for figure 4:

**Figure supplement 1.** Key interactions around the inserted residues in the D770-N771insNPG mutant as seen in the ligand-bound active conformation of the mutant (PDB ID: 4LRM), and in the homology model of the Src-like inactive conformation that we built.

**Figure supplement 2.** Simple moving averaged time series of the interactions of R776/9 with D770 and A767 over the course of the unbiased simulations starting from the monomeric active (top row) and Src-like inactive (bottom row) conformations.

**Figure supplement 3.** Amplification of the interlobal separation due to the D770-N771insNPG insertion.

In all the conformations that correspond to the global minimum of the reported free energy surface, the A-loop is predominately found in the extended, active-like conformation regardless of the order or position of the αC-helix. Inspection of the structures in this basin suggests that the A-loop is kept in the active conformation through the interaction of E762 with K863 of the nine sheet.

Interestingly, in the unbiased simulations of the mutant, as well as in a relatively populated cluster within the broad basin seen in the PTmetaD simulations, the insertion seems to amplify the N/C-lobe separation. The same N/C-lobe separation was seen also in the activator kinase in the asymmetric dimer (*Figure 1—figure supplement 4*). The role of this amplification, which is also seen in the A763-Y764insFQEA as it will be discussed later, is not yet clear, but it may be related to the Hsp90 recruitment and/or the binding kinetics of the substrate/inhibitors.

## A763-Y764insFQEA populates active-like states and semi-closed elongated states

When studied in vitro, the EGFR Ex20Ins variant A763-Y764insFQEA was the only EGFR Ex20Ins-harboring cell line inhibited by erlotinib at concentrations less than 0.1 μM, while other EGFR Ex20Ins, such as the D770-N771insNPG, had a reduced affinity and sensitivity to EGFR TKIs (*Yasuda et al., 2013*; *Hirano et al., 2018*). From the homology model of the A763-Y764insFQEA mutant that we built, which was based on the active conformation of the wild-type, it is not immediately clear how the mutation exerts its effect. Mutagenesis studies indicate that this insertion extends the αC-helix toward the N-terminal direction while the glutamic acid that is inserted through the mutation assumes the role of E762 in the WT-EGFR (*Yasuda et al., 2013*). The insertion leads to the replacement of I759 with alanine that has a shorter side chain. Again, the amino acid at position 759 is involved in hydrophobic interactions around L858 (*Figure 1—figure supplement 2*), which have been speculated to be essential for stabilizing the Src-like inactive state. Therefore, based on the homology model, it is tempting to think that the mutation has an activating effect by disrupting this hydrophobic network, which is found in the Src-like inactive conformation, and making the inactive conformation less energetically favorable.

In the case of the unbiased simulations that we initiated from the monomeric Src-like inactive conformation, the αC-helix remained in the αC-out conformation throughout the simulation (*Figure 1—figure supplement 2*), and the overall fold of the kinase was maintained with a noticeable increase in the interlobal distance. On the other hand, in the simulations that we initiated from the active conformation, the A763-Y764insFQEA was the only mutant in which the αC-helix transitioned back to the αC-in conformation several times after it visited the αC-out conformation (*Figure 1—figure supplement 2*). This unique behavior implies that the intrinsic tendency of the A750-N756 segment of the N-terminal of the αC-helix to fold that we see in the simulation may restrict the conformational flexibility of the αC-helix and favor αC-in conformations.

The deepest minimum in the free energy landscape of A763-Y764insFQEA – (CV1, CV2) = (−0.7, 0.3) – corresponds to an ensemble where the A-loop of the mutant samples semi-closed conformations, but unlike the WT, the αC-helix is stabilized in an αC-in conformation, where the K745-E762 salt bridge is formed (*Figure 5*, basin γ1). Interestingly, this ensemble contains several conformations in which the N-lobe has separated from the C-lobe, similar to the behavior seen in the D770-N771insNPG. In the case of the A763-Y764insFQEA, the activity of the mutant has been found to be sensitive to Hsp90 inhibitors, suggesting that this mutant is dependent on Hsp90 for stability and downstream-signaling (*Jorge et al., 2018*). As the function of this mutant has been shown to be chaperone-dependent (*Jorge et al., 2018*), the biological relevance of the observed elongated states with additional space between the two lobes might be to facilitate the recruitment of the Hsp90 chaperone.

In the second main minimum – (CV1, CV2) = (1.4, 0.1) – A763-Y764insFQEA samples inactive conformations where the two-turn helix on the A-loop is formed and stabilized by interactions of K860 with the inserted glutamic acid, and the shifted E762 and E758, as well as of interaction of L858 with the inserted F764 (*Figure 5*, basin γ2).

Notably, the fully active state is populated, albeit there is a small thermodynamic penalty of 1–2 kcal mol$^{-1}$ with respect to the inactive state. In the representative structure of this basin – (CV1, CV2) = (−0.45, 0.55) – the K745-E762(inserted Glu) salt bridge is formed, and the C is in the αC-in conformation (*Figure 5*, basin γ3). At the same time, the A-loop assumes an extended, active-like conformation.

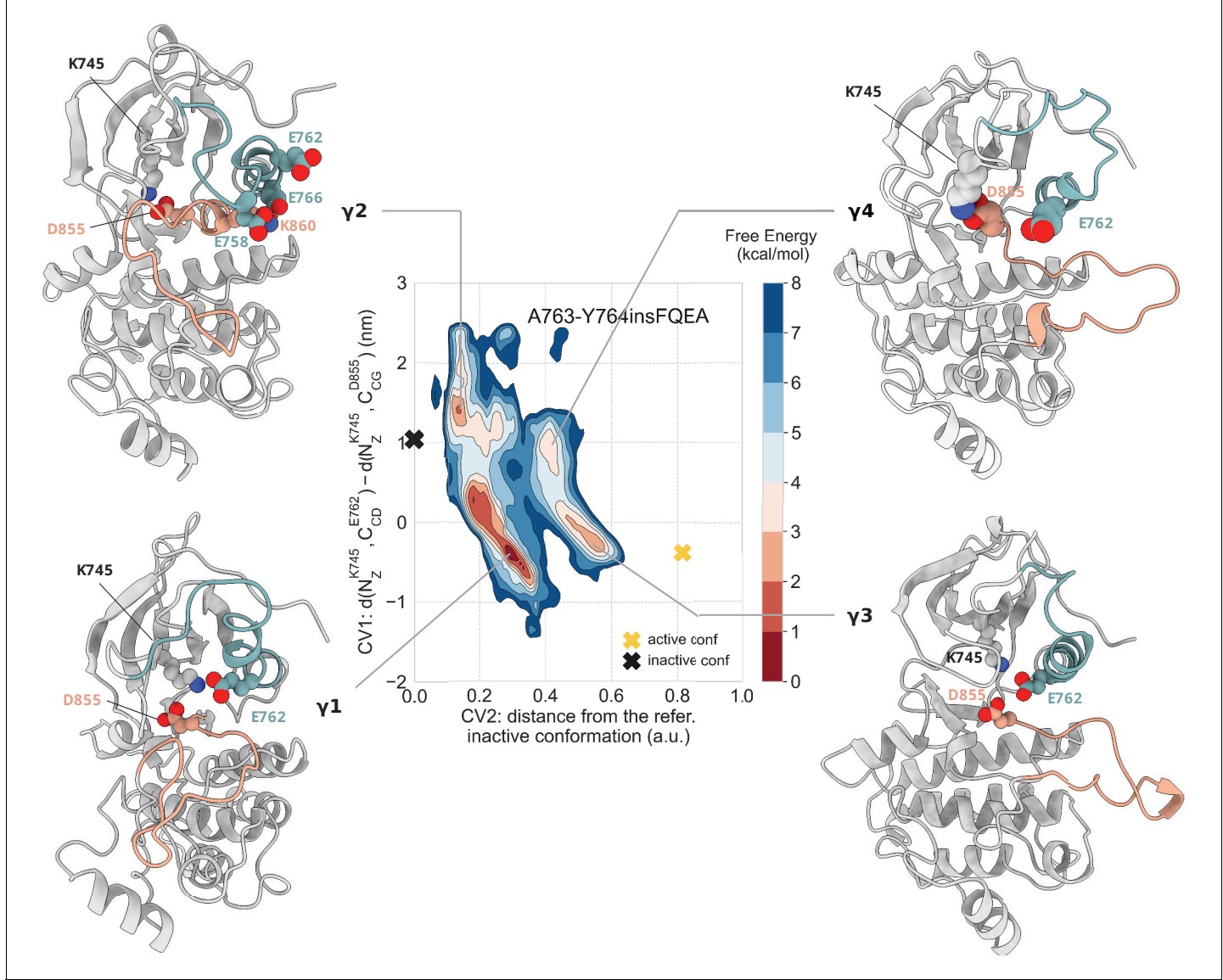

**Figure 5.** Free energy surface of the A763-Y764insFQEA mutant as a function of CV1 and CV2. Representative structures of the main conformations found in the free energy minima are depicted in cartoon representation. A yellow cross indicates the position of the active conformation (homology model) in the explored CV space, while a black cross indicates the position of the Src-like inactive conformation (homology model) as reference. The online version of this article includes the following figure supplement(s) for figure 5:

**Figure supplement 1.** Side and top view of the WT and A763-Y764insFQEA asymmetric dimer interface around the αC-helix (receiver monomer) and αH-helix (activator monomer).

Higher in energy we found an ensemble of conformations which has structural features from both basins 2 and 3. In this basin (*Figure 5*, basin γ4), the A-loop of A763-Y764insFQEA adopts the extended conformation seen in the active state, similar to basin 3, while the K745-E762 salt bridge is broken and replaced by the K745-D855. In this ensemble, the N-terminal end of the αC-helix is highly disordered. Although high in energy, this metastable state seems to be important for the transition to the active state, since the minimum energy path for this transition passes through it (*Figure 7—figure supplement 3*).

In the *apo*, monomeric form, the insertion results in an extension of the 3 C loop, which was expected to increase the mobility of the αC-helix. Instead, in both the unbiased and PTmetaD simulations that we performed, this extension seems to provide structural stability to the αC-helix. Even though the αC-helix samples disordered states in all the basins, in the dominant conformations of

these basins it is highly helical, suggesting that the mutation may also activate the receptor by suppressing the disorder of the αC-helix that is important for the formation of the asymmetric homodimer and downstream signaling. Moreover, looking at the position of the αC-helix in the two monomers of the asymmetric dimer, we see that, unlike the WT, the αC-helix is kept predominately in the αC-in conformation in both monomers (*Figure 1—figure supplement 4*). Interestingly, although the αC-helix of both monomers samples occasionally αC-out conformations, it transitions back to the αC-in conformation, even in the case of the donor kinase, where there is nothing to push the αC-helix back. We were not able to associate this behavior of the donor kinase with the mutation on the acceptor kinase or find an allosteric communication between the dimerization interface and the αC-helix. Nevertheless, since simulations of the monomeric state of A763-Y764insFQEA-EGFR shows that the αC-helix transitions naturally from the αC-out to the αC-in (*Figure 1—figure supplement 1*), we reason that the more frequent sampling of the αC-in conformation in the case of the donor kinase is most likely mutation-driven rather than dimerization-driven.

## ΔELREA populates αC-out conformations

EGFR mutants harboring ΔELREA show increased activity, comparable to the activity seen in L858R mutants and much higher than the one seen in the WT (*Foster et al., 2016*). Foster et al. showed that the deletion of five amino acids at the 3 C loop is optimal for kinase activation as this deletion is expected to prevent αC-helix from adopting an αC-out conformation (*Foster et al., 2016*). Moreover, it has been recently shown that ΔELREA EGFR is still oncogenic even in the absence of asymmetric homodimerization (*Cho et al., 2013*), unlike the wild-type and L858R EGFR, which acquire their oncogenic properties following asymmetric dimerization. However, we note that heterodimerization of the ΔELREA with other members of the ERBB family might play a role.

Modeling of the ΔELREA deletion in the active conformation suggests that the deletion does not significantly perturb the overall kinase domain. Inspection of the region around the mutation indicates that the deletion can be accommodated in the active, αC-in conformation by repositioning of residues $^{753}$PKAN$^{756}$, which form the initial N-terminal turn of the αC-helix in the WT-EGFR after the three-strand. On the other hand, modeling of the Src-like inactive conformation suggests that unfolding of this turn at the N-terminal end of the αC-helix is necessary for it to adopt the characteristic αC-out conformation. Long unbiased molecular dynamics simulations by Shan et al. suggest that the ΔELREA mutation stabilizes the αC-in active conformation relative to the Src-like inactive conformation, but the mutation does not prevent αC-helix from sampling the αC-out conformation (*Shan et al., 2012*). In our unbiased simulations, we confirm that the αC-out is a stable conformation in terms of the structural integrity of the two-turn helix on the A-loop, the αC-helix and the overall folding of the receptor.

The reconstructed free energy landscape confirms this observation, as the αC-helix is found in the αC-out conformation in one of deepest free energy minima – (CV1, CV2) = (1.4, 0.1) – the characteristic two-turn helix on the A-loop of the Src-like inactive conformation is formed, and the αC-helix has shifted to adopt the αC-out conformation (*Figure 6*, basin β2). The deleted L747, which on the WT contributes to the stabilization of the two-turn helix through a hydrophobic network between I759, L861, and L858 (*Figure 1—figure supplement 3*), is replaced by P753 (*Figure 6*, basin β2) and, therefore, the stability of the Src-like inactive conformation is not compromised despite the deletion.

In the second free energy minimum – (CV1, CV2) = (1.6, 0.45) – ΔELREA sampled mainly active-like conformations in which the C was again in the αC-out conformation, but the A-loop assumed extended conformations (*Figure 6*, basin β1) similar to those seen in the unbiased simulations of the active WT-EGFR. Interestingly, the A-loop in this ensemble also visited conformations in which the middle section of the A-loop formed a three-turn helix, reminiscent of an extended helix in the A-loop of MPSK1 kinase (*Figure 6—figure supplement 1*, PDB ID: 2BUJ). This conformation is intriguing because the helix puts Y869, a well-known phosphorylation site of EGFR kinase, in an exposed position along with a group of glutamate residues (E865, E866, E868, and E872), which interact with K754 of the unfolded N-terminal end of the αC-helix (*Figure 6*, basin β1). Interestingly, in a basin between 1 and 2, this conformation that is less populated in basin two becomes dominant and we see Y869 interacting with K754 and E762 with K860 (*Figure 6*, basin β4). Moreover, the minimum energy path connecting the Src-like inactive to active conformation passes through this basin (*Figure 7—figure supplement 3*), which suggests that interactions of the unfolded N-terminal end

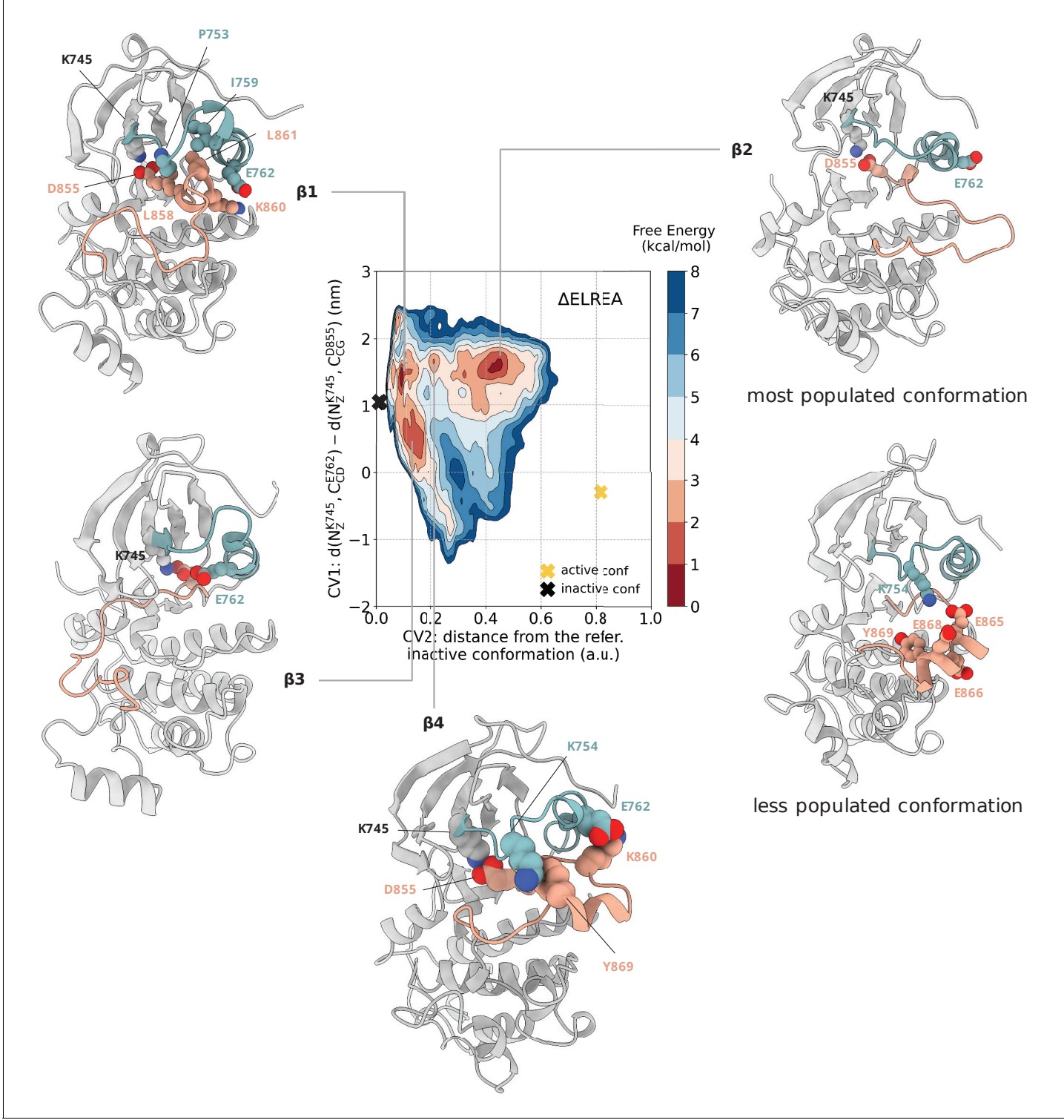

**Figure 6.** Free energy surface of the ΔELREA mutant as a function of CV1 and CV2. Representative structures of the main conformations found in the free energy minima are depicted in cartoon representation. A yellow cross indicates the position of the active conformation (homology model) in the explored CV space, while a black cross indicates the position of the Src-like inactive conformation (homology model) as reference.

The online version of this article includes the following figure supplement(s) for figure 6:

**Figure supplement 1.** Superposition of a representative structure from an ensemble of conformations explored by the ΔELREA mutant (cyan) with MPSK1 kinase (beige, PDB ID 2BUJ).

**Figure supplement 2.** ΔELREA interface interactions ΔELREA interface interactions.

of the αC-helix, which is now the 3 C loop, with the A-loop in the ΔELREA mutant should be important. This helical conformation of the activation loop might be involved in the phosphorylation of Y869. A similar conformation has been described in *Shan et al., 2013* as an intermediate state of the active to Src-like inactive transition of the WT.

The third metastable minimum, slightly higher in energy than the other two, is very broad and consists of a divergent ensemble in which ΔELREA samples a range of conformations most of which are 'substrate-competitive' like (*Figure 1—figure supplement 1*), that is neither the two-turn helix nor the nine strand on the A-loop is stable and the residues of the DFG motif are in the DFG-in orientation (*Figure 6*, basin β3). In particular, in this ensemble, the A-loop points to the opposite direction with respect to the one seen in the active conformation, as seen in many kinases like the inactive conformation of the AuroraA kinase (PDB ID: 2WTV).

Although the ΔELREA is one of the most frequent mutations in NSCLC, the effect of ΔELREA on the dimerization ability of the ΔELREA-EGFR is poorly understood. Our simulations of the ΔELREA-EGFR in the asymmetric dimer do not show any significant alteration on the interface of the mutant dimer with respect to the WT, apart from the marginal stabilization of the inter-monomeric salt bridges K944-E962 and K752-E958 (*Figure 6—figure supplement 2*). We anticipate, however, that suppression of the intrinsic disorder of the αC-helix of the monomeric mutant that we see in all of the explored conformations during the PTmetaD simulations should promote the formation of active asymmetric dimers.

## Discussion

In light of the clinical importance of EGFR kinase domain mutations in lung cancer, understanding the unique properties of mutant kinases is crucial to drug development and lung cancer biology and provides a good opportunity to understand better the biophysical mechanisms regulating receptor kinase activation. The design of conformation- and mutation-specific kinase inhibitors to target a diversity of cancer-causing mutations is of paramount importance to achieve the optimal therapeutic outcome for cancer patients and represents an unmet clinical need in precision oncology. A better understanding of how oncogenic mutations affect, at a molecular level, the structure and dynamics of the receptor with respect to the WT may potentially lead to personalized cancer treatment tailored to the genetic profile of each cancer patient. Here, we have presented the effect of four such mutations, namely the L858R, D770-N771insNPG, ΔELREA, and A763-Y764insFQEA, all of which have been shown to increase the catalytic activity of EGFR in vitro (*Zhang et al., 2006*; *Foster et al., 2016*; *Yasuda et al., 2013*) and continuously signal the pathway, which stimulates cancer cell growth.

It has been well established that the activation of WT-EGFR is driven by ligand-induced dimerization or multimerization (*Zanetti-Domingues et al., 2018*; *Lemmon and Schlessinger, 2010*). In agreement with previous studies (*Shan et al., 2013*; *Shan et al., 2012*; *Sutto and Gervasio, 2013*), we observe that the active conformation of the WT-EGFR, in the absence of phosphorylation and ATP, is energetically unfavorable and infrequently sampled. The absence of a low-energy αC-in active conformation in which the catalytically important KE salt bridge between K745 and E762 is maintained suggests that dimerization is important to stabilize this conformation fully. Indeed, in our unbiased simulations of asymmetric EGFR homodimers, we see that the αC-helix is kept in the αC-in conformation predominately on the receiver monomer both of the WT and of the mutants (*Figure 1—figure supplement 4*). Dimerization seems to suppress the intrinsic disorder at the receiver interface stabilizing the active conformation of the receiver kinase, as this is reflected on the increased stability of the KE salt bridge in the receiver kinase, but not in the activator. However, the αC-helix of the receiver is still flexible enough to sample occasionally αC-out conformations in most of the asymmetric dimers, albeit in lower frequency than the monomeric kinase or when it acts as a donor, which suggests that the binding of ATP might still be essential for the αC-helix to be locked fully in the αC-in conformation.

It should be noted that apart from the dimerization, which is affected by the order-disorder transitions of the C helix, the binding of the substrate and the phosphorylation of the A-loop and the C-terminal segment will also play a role in the relative population of active and inactive conformations (*Kim et al., 2012*; *Kovacs et al., 2015*). For instance, the phosphorylation of the A-loop, albeit not strictly necessary to propagate the signal downstream (*Zhang et al., 2006*), affects the flexibility

of the loop (*Shan et al., 2012*), further stabilizing the active state and affecting the catalytic efficiency both directly and indirectly.

We reasoned that a collective motion that might be more directly linked to the steady state catalytic activity might be that of the hinge region, which has been linked to the phosphorylation of the substrate (*Pucheta-Martínez et al., 2016*), the active to inactive transition (*Shan et al., 2012*), and also the release of the products. The unbiased simulations of the different mutants in the monomeric and homodimeric form show that, the monomer explores more 'open' hinge conformations with respect to the receiver kinase of the dimer, which is in agreement with the activating role of dimerization. Interestingly, two of the mutants (ΔELREA and L858R) explore more 'closed' hinge conformations in the dimer (*Figure 7—figure supplement 4A,B*).The same trend is clearly visible in the reprojection of FE with respect to the interlobal distance, where the position of the global minimum follows the same order as the catalytic efficiency (*Figure 7—figure supplement 4C*). Although we can only speculate about this, it is likely that this compressed kinase seen in ΔELREA and L858R might be linked to slow release of the substrate/ligands.

In the global minimum of the free energy landscape of the WT-EGFR, the A-loop has adopted a semi-closed conformation that precludes the ATP-binding, and the αC-helix has adopted the αC-out conformation. The N-terminal end of it is partially unfolded in conformations that correspond to local minima, which is in line with the disordered nature of the αC-helix (*Shan et al., 2012*).

Light scattering and native gel electrophoresis have suggested that, unlike the WT-EGFR, which is found predominately in monomers in solution, the L858R EGFR preferentially forms dimers and oligomers (*Shan et al., 2012*). Earlier studies have shown that L858R promotes dimerization by suppressing the intrinsic disorder of the αC-helix (*Shan et al., 2012*; *Sutto and Gervasio, 2013*), which is an integral component of the dimerization interface, while it also promotes the active state. Our simulations corroborate this combined mechanism of action. The deepest free energy minimum corresponds to active conformations in which the interaction of R858 with D837 and D855 disfavors the formation of the characteristic two-turn helix on the A-loop of the Src-like inactive conformation, while the interaction of R858 with E758 and E762 represses the disorder in the αC-helix region, thus favoring the formation of asymmetric homodimers.

In-frame Ex20Ins are a subcategory of EGFR mutations, most of which are found in the αC-β4 loop that connects the αC-helix to the β4-strand of the kinase domain. TKIs (afatinib, lapatinib, neratinib, dacomitinib) targeting Ex20Ins EGFR have been shown to have limited activity in patients with EGFR-Ex20Ins-mutant tumors, with exception of the covalent inhibitor poziotinib, which has been found to demonstrate greater activity in vitro (*Yasuda et al., 2013*; *Ruan and Kannan, 2018*; *Robichaux et al., 2018*) and in patient-derived xenograft models of EGFR Ex20Ins mutant NSCLC and in genetically engineered mouse models of NSCLC (*Robichaux et al., 2018*). According to our simulations, we reason that the amplification of the lobe separation that we observed in both the unbiased and PTmetaD simulations could explain the low sensitivity of D770-N771insNPG-EGFR to TKIs, as this motion is expected to affect the residence time of the TKIs in the binding site. Unlike previous suggestions, according to which the inserted residues would prevent the receptor from adopting an αC-out conformation (*Lemmon and Schlessinger, 2010*), we sample such conformations, in which though the A-loop is maintained in an extended, active conformation. Interestingly, BDTX-189, a recently developed inhibitor that targets Ex20Ins mutations, exhibited potent tumor growth inhibition and tumor regression in preclinical models (*Hamilton et al., 2020*). The scaffold of BDTX-189 is similar to that of the lapatinib analogue neratinib, which suggests that BDTX-189 probably binds αC-out conformations in accordance with the low energy of such states on the free energy landscape of D770-N771insNPG. It is possible that the mutation decouples the N/C-lobe motions and subsequently the coupling between the αC-helix and the A-loop –a coupling that is important for the transition from the active to the inactive state (*Huse and Kuriyan, 2002*) – and makes the transition to the inactive conformation less favorable.

Unlike the majority of EGFR Ex20Ins (including the relatively rare D770-N771insNPG), which appear to be relatively insensitive to early EGFR TKI's, recent data show that the A763-Y764insFQEA insertion has greater sensitivity to agents such as gefitinib, erlotinib, and osimertinib (*Yasuda et al., 2013*; *Floc'h et al., 2018*). Our simulations show that the most stable conformation of the *apo* A763-Y764insFQEA corresponds to an αC-in conformation in which, however, the A-loop is not fully extended as seen in the active conformation. Interestingly, the fully active conformation that is not favorable in the *apo* WT-EGFR, appears as a relatively low energy secondary minimum in the FES of

the A763-Y764insFQEA. What is more, after simulation of a homology model of the mutation in the asymmetric dimer, we anticipate the mutation to also promote the formation of such dimers, which would explain the higher activity of A763-Y764insFQEA-EGFR compared to D770-N771insNPG-EGFR. Specifically, the introduced F760 of A763-Y764insFQEA is accommodated in the hydrophobic cleft formed by V952, M949, and K953, while K757 of the receiver kinase in the WT is replaced by D761 in the A763-Y764insFQEA, which can interact with K953 of the activator kinase (*Figure 5—figure supplement 1A*). Moreover, the introduction of the insertion and the consequent extension of the αC-helix by one turn brings K757 and K754 in such a position that allows them to interact more efficiently with the side chains of E967 of the αI-helix, and of D960 and D958 of the αH-αI loop, respectively (*Figure 5—figure supplement 1B*). Notably, these interactions cannot not occur in the WT.

Through PTmetaD simulations, we explored the differences between the conformational landscapes of EGFR Ex20Ins and the WT, L858R, and ΔELREA mutants. Such differences are not evident from the available crystal structures where the residues of the ATP pocket appear to be identical among the different structures. We believe that the conformational ensembles from the PTmetaD simulations of the EGFR Ex20Ins with the more extended conformations that lead to a more open active site can be used to identify novel inhibitors with increased selectivity against the mutant EGFR. We should note that in such drug design efforts, using the appropriate conformation is of great importance. For example, we can already see that targeting the L858R-EGFR with αC-out inhibitors, like lapatinib, would lead to steric clashes that are not present in the pocket of an Ex20Ins EGFR like A763-Y764insFQEA, while inhibitors like gefitinib can be accommodated better in the tighter pocket of L858R (*Figure 7—figure supplement 5*).

It has been postulated that the β3-αC deletion forces the αC-helix into the αC-in position promoting the kinase activation and decreasing the sensitivity to αC-out inhibitors, like lapatinib (*Foster et al., 2016*). The free energy landscape of the ΔELREA mutation, however, suggests that this notion needs to be revisited. The most stable conformations of the unphosphorylated monomeric form correspond to minima in which the C-terminal end of the β3-strand unfolds, giving the αC-helix the necessary flexibility to adopt αC-out conformations. However, both the unbiased simulations and the conformations within all the energy basins show that the ΔELREA deletion almost completely suppresses the local disorder at the N-terminal end of the αC-helix region, which in turn is expected to favor the formation of asymmetric dimers that could explain the increased activity of ΔELREA EGFR with respect to the WT. Moreover, simulations of the binding of lapatinib to the WT-EGFR have suggested that the slow kinetics of lapatinib's binding are probably associated with the transition to a conformation in which the αC-helix is partially unfolded (*Shan et al., 2012*). It is, thus, tempting to speculate that the absence of such intrinsically disordered conformations in the free energy landscape of ΔELREA could explain the decreased sensitivity of ΔELREA-EGFR to lapatinib.

Our study reveals intricate differences between EGFR mutations, even between mutations that occur within the same exon. In light of the present results, we can rationalize the activation effect of these mutations in an atomic level resolution. Inhibiting the activity of abnormal EGFR proteins, as well as other proteins in the pathway, can interrupt this signaling pathway that causes cancer cells to grow.

## Materials and methods

### Protein structure preparation - monomeric EGFR

The crystal structures for the active, unphosphorylated kinase domain of the wild-type (PDB ID: 2GS2) and mutant (PDB ID: 2ITV for the L858R mutant, and PDB ID: 4LRM for the D770-N771insNPG mutant) EGFR were retrieved from the Protein Data Bank and the used sequence comprises the interval L703–Q976 in our notation, L67–Q952 in the PDB notation. The amino acid numbering convention adopted here includes the 24-residue long membrane-targeting signaling peptide that is deleted in the mature protein.

The missing residues of the A-loop in 2GS2 were added using the software Modeller, according to the respective Uniprot sequence. Any co-crystallized ligand present in the crystal structures was removed. The structure of the active WT was used then as a template to model the missing residues in the structures of the L858R and D770-N771insNPG mutants. In the absence of crystallographic

data regarding the structures of the active A763-Y764insFQEA, and ΔELREA mutations, we used the structure of the WT to introduce the mutations using the automodel functionality of MODELLER (*Sali and Blundell, 1993*). In the case of the A763- Y764insFQEA mutant EGFR, the four-residue FQEA insertion occurs in the middle of the αC-helix. The one turn of a helix that these residues form was modeled such that it shifts the register of adjacent residues in the helix toward the N-terminal direction, in accordance with the mutagenesis studies performed by *Yasuda et al., 2013* to determine the direction of the shift.

Since none of the studied mutants has been crystallized in an inactive conformation, we used the crystal structure of the unphosphorylated WT-EGFR in the Src-like inactive conformation (PDB ID: 2GS7) to create models of the Src-like inactive states of the mutants. The mutations were introduced using MODELLER (*Sali and Blundell, 1993*).

MD simulations combined with experimental data have shown that the flip of the DFG motif in many protein kinases, including the kinase domain of EGFR, is dependent to the protonation state of the Asp residue of the DFG motif where protonation promotes the flip and favors the DFG-out conformation (*Shan et al., 2013*; *Shan et al., 2009*; *Sultan et al., 2017*). In all the structures that were used in the simulations that are described below, the Asp was unprotonated and, therefore, the DFG-flip was not sampled in any of the simulations described here.

## Protein structure preparation - homodimeric EGFR

The *apo*, unphosphorylated WT-EGFR monomers were assembled into asymmetric and head-to-head symmetric homodimers based on the crystal packing of the monomers observed in PDB entries 2GS6 and 5CNO respectively (*Zhang et al., 2006*; *Kovacs et al., 2015*). In the asymmetric dimeric configuration, the kinase domain of each monomer is found in the active conformation, while in the head-to-head symmetric homodimer the kinase domains are found in the Src-like inactive conformation.

Out the four mutants that we simulated here, available crystal structures of mutant dimers exist only for the L858R EGFR (PDB ID: 2ITV) and D770-N771insNPG-EGFR (PDB ID: 4LRM), the monomers of which form active, asymmetric assemblies in their crystal structures. Given the activating nature of all the studied mutants, and in the absence of crystallographic data for symmetric dimers for any of them, we simulated the mutants only in the asymmetric dimers. The A763-Y764insFQEA and ΔELREA were modeled to the asymmetric dimer using the WT-EGFR as a template using the software MODELLER (*Sali and Blundell, 1993*).

## Unbiased MD simulations details - monomeric EGFR

Prior to any simulation, the protonation state of each residue was calculated using PROPKA3.0 server (*Li et al., 2005*) and correspond to pH 7. Then, each system was enclosed in a dodecahedron box with periodic boundary conditions and solvated with TIP4PD water molecules, while $Na^+$ and $Cl^-$ ions were added to reach neutrality, and the final NaCl concentration of 150 mM. The MD simulations were performed using the GROMACS 2018.3 simulation package (*Abraham et al., 2015*) patched with the PLUMED 2.4.1 plug-in (*Tribello et al., 2014*). For all simulations, we used a99SB-disp protein force field with its modified TIP4PD water model (*Robustelli et al., 2018*). The energy of all systems was minimized using the steepest descent integrator and the solvated systems were equilibrated afterwards in the canonical (NVT) ensemble for 5 ns, using a Berendsen barostat (*Berendsen et al., 1984*) at 1 bar with initial velocities sampled from the Boltzmann distribution at 300 K. The temperature was kept constant at 300 K by a velocity-rescale thermostat (*Bussi et al., 2007*) and a time step of 2 fs. The long-range electrostatics were calculated by the particle mesh Ewald algorithm, with Fourier spacing of 0.16 nm, combined with a switching function for the direct space between 0.8 and 1.0 nm for better energy conservation. The systems were equilibrated for additional 10 ns in the isothermal-isobaric (NPT) ensemble prior to the production run applying position constraints to the protein (with a restraint spring constant of 1 kJ $mol^{-1}nm^{-2}$). An unconstrained 1000 ns production run in the NPT ensemble, coupled with a velocity-rescale thermostat (*Bussi et al., 2007*) at 300 K and a Parinello-Rhaman barostat (*Parrinello and Rahman, 1981*) at 1 bar, was carried out for each of the systems starting independently from the active and Src-like inactive conformation (*Table 1*).

## Unbiased MD simulations details - homodimeric EGFR

For the unbiased simulations of the asymmetric dimers, a similar protocol to the monomeric EGFR was followed for consistency. In particular, the protonation states of the residues at pH 7 were determined by PROPKA3.0 (*Li et al., 2005*), which left all the residues in their usual charge states. Then, each system was enclosed in a truncated dodecahedron box with periodic boundary conditions and solvated with TIP4PD water molecules, while $Na^+$ and $Cl^-$ ions were added to reach neutrality, and the final NaCl concentration of 0.15 M (the total number of atoms was ~230,000 per system). All molecular dynamics simulations were performed using a99SB-disp (*Robustelli et al., 2018*) protein force field in the NPT ensemble, keeping the temperature at 300 K with a velocity-rescale thermostat (*Bussi et al., 2007*), and the pressure at 1 bar with a Parinello-Rahman barostat (*Parrinello and Rahman, 1981*). The long-range electrostatics were calculated by the particle mesh Ewald algorithm, with Fourier spacing of 0.16 nm, combined with a switching function for the direct space between 0.8 and 1.0 nm for better energy conservation. The systems were equilibrated for additional 10 ns in the NPT ensemble prior to the production run applying position constraints to the protein (with a restraint spring constant of 1 kJ mol$^{-1}$ nm$^{-2}$). An unconstrained 4000 ns production run in the NPT ensemble was carried out for each of the systems (*Table 2*).

## Metadynamics simulations details - monomeric EGFR

A preliminary PTmetaD simulation in the well-tempered ensemble was performed for each of system using eleven replicas at increasing temperatures (300, 305, 310, 318, 326, 335, 344, 354, 363, 375, and 382 K), in which only the potential energy was biased so that to increase its fluctuation and reach a target exchange rate between neighboring replicas of 15%. To ensure that all meaningful conformations would be sampled over the course of the PTmetaD simulations, three out of the 11n replicas were started with the WT or mutant EGFR in the Src-like inactive conformation, whereas in the rest the kinase was in the active conformation. Once the desired exchange rate between the replicas was reached, the simulations were interrupted and the deposited energy was saved and used as an initial bias during the production PTmetaD simulations.

After this preliminary run, for the production stage, we run PTmetaD in the well-tempered ensemble for each system using the eleven replicas (300 to 382 K), where we kept biasing the potential energy by depositing gaussians of height, $W_0 = 1$ kJ mol$^{-1}$, and width, $\sigma_{PotEnergy} = 100$ kJ mol$^{-1}$ to maintain a good exchange rate despite the big temperature separation. An exchange between adjacent replicas was attempted every 2 ps. The deposited Gaussians from the preliminary well-tempered run were loaded in the PTmetaD simulation to maintain the exchange rate between the replicas to 15%. A Gaussian was deposited in the collective variable space every 1 ps with $W = W_0 e^{-\frac{V(s,t)}{(f-1)T}}$, where $W_0 = 4$ kJ mol$^{-1}$ is the initial height, T is the temperature of each replica, $f = 15$ is the bias factor, and V(s, t) is the bias potential at time t and CV value s.

The following three collective variables were biased simultaneously during the production stage: the difference between atomic distances $CV1 = d(N_Z^{K745}, C_\delta^{E762}) - d(N_Z^{K745}, C_\gamma^{D855})$; the distance in contact map space with respect to the inactive conformation $CV2(R) = \frac{1}{N}\sum_{\gamma \in \Gamma}(D_\gamma(R) - D_\gamma(R_{inactive}))^2$; the distance in contact map space with respect to the active conformation

**Table 2.** Summary of the unbiased simulations of the homodimeric EGFR.
The reported times correspond to the simulation time of each independent unbiased simulation.

| System | Starting conformation | Total simulation time (ns) |
|---|---|---|
| WT | asymmetric dimer | 4000 |
| | symmetric dimer | 4000 |
| L858R | asymmetric dimer | 4000 |
| A763-Y764insFQEA | asymmetric dimer | 4000 |
| D770-N771insNPG | asymmetric dimer | 4000 |
| ΔELREA | asymmetric dimer | 4000 |

$CV3(R) = \frac{1}{N}\sum_{\gamma \in \Gamma}(D_\gamma(R) - D_\gamma(R_{active}))^2$. $D_\gamma(R)$ is a sigmoidal function that measures the degree of formation of the contact $\gamma$ in the structure R and is defined as $D_\gamma(R) = w_\gamma \frac{(r_\gamma/r_\gamma^0)^n}{(r_\gamma/r_\gamma^0)^m}$, where $r_\gamma$ is the contact distance in the structure $R$, $r_\gamma^0$ is the contact distance in either the reference inactive or active conformation depending if $\gamma$ is a contact specific to the former or latter conformation, $w_\gamma$ is the weight of the contact and is set to one for regular contacts and three for salt bridges, N is a normalization constant, $n = 6$, and $m = 10$. Given the previous success of these three CVs to probe the effect of oncogenic mutations on the free energy landscape of EGFR's kinase domain, here, we used the same set of contacts for the contact map distance of CV2 and CV3 as the ones used by Sutto et al. for the WT (*Sutto and Gervasio, 2013*). The set of contacts includes only those pairs that are able to discriminate between the two structures, that is, pairs of atoms that describe contacts, which are formed in the active conformation but are broken in the Src-like inactive and vice-versa.

To further assess whether these contact maps and their adaptation from Sutto et al. were indeed able to describe the transition of interest, prior to the PTmetaD simulations, we ran a set of steered-MD (SMD) simulations, where we steered the active to Src-like inactive transition along the contact map space. In these exploratory SMD simulations, the decrease of the root-mean-square deviation (RMSD) with respect to the Src-like inactive conformation when we start from the active conformation and steer toward the Src-like conformation (*Figure 7*) as well as the satisfactory superposition of the final SMD conformation to the Src-like inactive conformation confirmed the adequacy of these contact maps, which were then used in the PTmetaD.

The PTmetaD simulations were run until the free energy in the bidimensional projections and in the monodimensional projections did not change more than 2.5 kcal mol$^{-1}$ in the last 100 ns and diffusive behavior was reached for all CVs (*Figure 7—figure supplement 1*). This convergence criterion led to 1000 ns/replica long simulations for WT, A763-Y764insFQEA, and ΔLREA, and to 900 ns/replica long simulations for L858R and D770-N771insNPG.

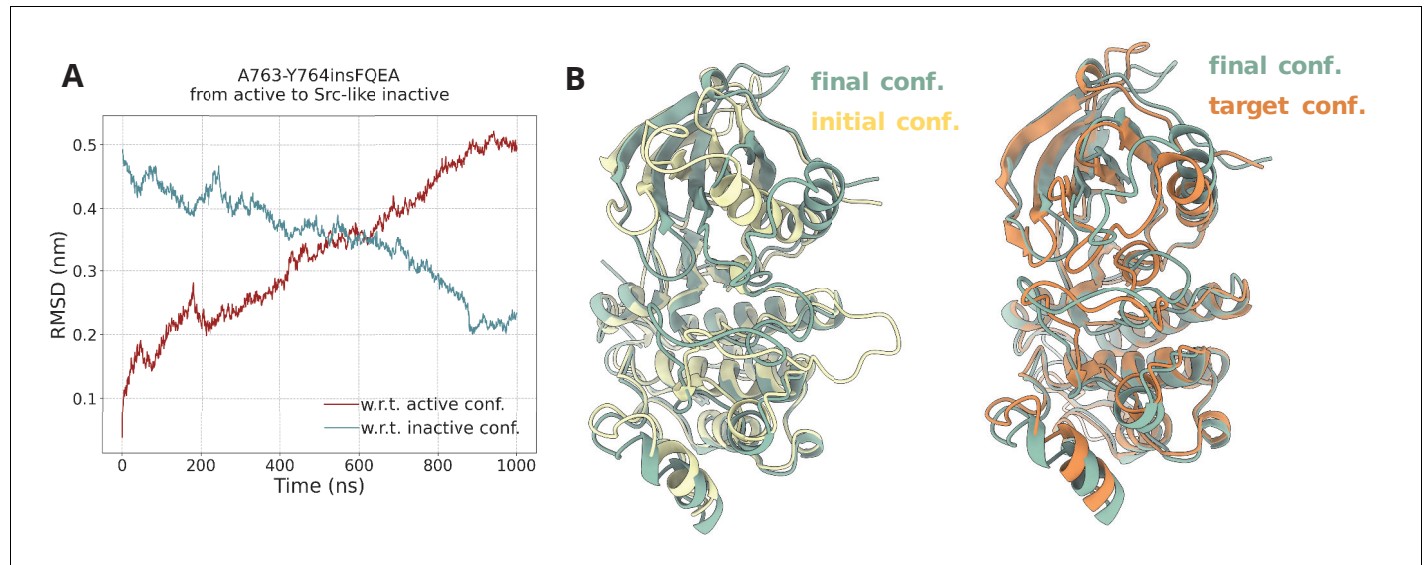

**Figure 7.** Steered MD simulations. (**A**) RMSD change with respect to the active and Src-like inactive conformation during the SMD simulation of the A763-Y764insFQEA EGFR. The SMD simulation was initiated from the active conformation (yellow) and streered toward the Src-like inactive conformation (orange). (**B**) Superposition of the final frame of the SMD simulation with the initial and target conformation. Same SMD simulations were run for each system prior to the PTmetaD to confirm that the designed contact maps can drive the active to Src-like inactive transition and vice-versa with similar results (data not shown).

The online version of this article includes the following figure supplement(s) for figure 7:

**Figure supplement 1.** PTmetaD simulations convergence check.

**Figure supplement 2.** Mapping of the CV spaced sampled during the unbiased simulations to the PTmetaD simulations.

**Figure supplement 3.** Minimum energy path.

**Figure supplement 4.** N/C-lobe separation.

**Figure supplement 5.** Fitting of gefitinib and lapatinib to L858R and A763-Y764insFQEA.

The input files for the PTmetaD simulations as well as the necessary files for the definition of the contacts that were used as CVs will be deposited to the public repository of the PLUMED consortium, PLUMED-NEST (*PLUMED Consortium, 2019*).

Replica exchange methods are generally based on the idea of sampling one 'cold' replica, from which the unbiased statistics can be extracted, plus a number of 'hot' replicas, whose only purpose is that of accelerating sampling. The 'hottest' replica should explore the space fast enough to overcome barriers for the process under investigation, whereas the intermediate replicas are necessarily introduced to bring the system smoothly from the 'hottest' ensemble to the 'coldest' ensemble. In the parallel tempering scheme that we used, the temperature is exploited to enhance the phase-space exploration within a replica exchange scheme, which ensures the correct sampling of the canonical ensemble for all the replicas. The free energy surface of each mutant was reconstructed by integrating the deposited bias during the simulation of the biologically relevant and lowest-temperature replica (T = 300 K), as required by the metadynamics algorithm.

To obtain a representative structure of each free energy basin, a clustering of the ensemble of conformations falling within each basin has been performed. The gromos algorithm of the *g_cluster* program from the GROMACS package has been used with the RMSD on the C atoms of the A-loop and the αC-helix residues (*Pan et al., 2014*) on the ensemble of the 300 K replica as the clustering metric with a cutoff of 0.2 nm. In the most populated cluster of each basin, the central structure, that is, the structure with the smallest distance to all of the other members of the cluster, has been picked as representative of the basin.

To map the position of the equivalent mutant structures on the contact map space of the WT, the distance between the atoms of each contact in the energy minimized structures of each mutant was compared to WT-reference structure and the degree of formation of each contact in the structure R was calculated using the sigmoid function that is described above. Deviation of the position of the atoms that define each contact in each mutant from the position of the equivalent atoms in the WT structure that was used as the reference, results in different positions of the crosses in the contact map space of the WT. On top of that, for the FES plots, the calculated contact-map distances were normalized and, therefore, mutants that explored larger portions of the contact-map space comparatively push the position of crosses further toward zero.

To trace the minimum energy path that connects the active to Src-like inactive conformation based on the calculated FES we used the MEPSAnd tool (*Marcos-Alcalde et al., 2020*) in a similar way to that described previously to characterize complex paths associated with biomolecular processes, including conformational transitions, and ligand (un)binding (*Berteotti et al., 2009*; *Bernetti et al., 2019*).

## Acknowledgements

IG is funded by Astra Zeneca-EPSRC case studentship awarded to FLG and SH. HEC-BioSim (EP/R029407/1), PRACE (Barcelona Supercomputing Center, Project BCV-2019-3-0010) and the Swiss National Supercomputing Centre (CSCS) (Project 280 and S847) are acknowledged for their generous allocation of supercomputer time.

## Additional information

### Competing interests

Luca Carlino, Richard A Ward, Samantha J Hughes: is affiliated with AstraZeneca. The author has no financial interests to declare. The other authors declare that no competing interests exist.

### Funding

| Funder | Grant reference number | Author |
| --- | --- | --- |
| EPSRC | Astra Zeneca-EPSRC case studentship | Ioannis Galdadas<br>Shozeb Haider<br>Francesco Luigi Gervasio |
| AstraZeneca | AstraZeneca-EPSRC case | Ioannis Galdadas |

studentship

The funders had no role in study design, data collection and interpretation, or the decision to submit the work for publication.

## Author contributions
Ioannis Galdadas, Conceptualization, Data curation, Formal analysis, Investigation, Visualization, Methodology, Writing - original draft, Project administration; Luca Carlino, Formal analysis, Investigation, Writing - review and editing; Richard A Ward, Conceptualization, Writing - review and editing; Samantha J Hughes, Shozeb Haider, Conceptualization, Supervision, Methodology, Writing - original draft; Francesco Luigi Gervasio, Conceptualization, Supervision, Funding acquisition, Methodology, Writing - original draft, Project administration, Writing - review and editing

## Author ORCIDs
Ioannis Galdadas (iD) https://orcid.org/0000-0003-2136-9723
Richard A Ward (iD) https://orcid.org/0000-0002-2310-9477
Shozeb Haider (iD) http://orcid.org/0000-0003-2650-2925
Francesco Luigi Gervasio (iD) https://orcid.org/0000-0003-4831-5039

## Decision letter and Author response
Decision letter https://doi.org/10.7554/eLife.65824.sa1
Author response https://doi.org/10.7554/eLife.65824.sa2

# Additional files

## Supplementary files
• Transparent reporting form

## Data availability
All input data for the simulations ran and analysed during this study have been deposited to the public repository of the PLUMED consortium, PLUMED-NEST.

The following dataset was generated:

| Author(s) | Year | Dataset title | Dataset URL | Database and Identifier |
|---|---|---|---|---|
| Gervasio FL | 2021 | EGFR activating mutations mechanism | https://www.plumed-nest.org/eggs/21/027 | PLUMED-NEST, plumID:21.027 |

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
