## [Decision Letter]

**Acceptance summary:**

Using unbiased molecular dynamics simulations and metadynamics simulations, this work characterized the conformational landscapes of the wild-type EGFR kinase and a number of oncogenic mutants. It suggests that the effects and the underlying activation mechanisms of these mutations are varied. In particular, it suggests that the Exon20-deletion mutant tends to adopt a more open conformation, which may be a potentially important finding for drug discovery.

**Decision letter after peer review:**

Thank you for submitting your article "Structural basis of the effect of activating mutations on the EGF receptor" for consideration by *eLife*. Your article has been reviewed by 2 peer reviewers, and the evaluation has been overseen by a Reviewing Editor and Jonathan Cooper as the Senior Editor. The reviewers have opted to remain anonymous.

Summary:

The authors present a metadynamics study comparing the free energy landscape of epidermal growth factor receptor (EGFR) and highlight the unique ways by which oncogenic mutations alter EGFR free energy landscape. The work identified a number of intermediate states unique for some of the mutants, which may be exploited in drug discovery. The characterization of one specific mutant, the Exon 20 insertion, is particularly timely, as many EGFR inhibitor drugs are inactive to this mutant, and drug discovery targeting this mutant therefore remains a challenge.

The reviewers raised a number of important concerns that need to be addressed to make this manuscript suitable for publishing in *eLife*.

Essential Revisions:

1) The authors should compare the kinetics of inactive-active transition for all four mutants and map the lowest energy path from the inactive to active state. If the change in free energy of transition in these mutants correlate to their known fold-change in catalytic activity, this would provide additional support for their model.

2) The observation that the Exon 20 mutants furnish a more open active site is perhaps the most direct and interesting result of this study. This should be further explored. For example, the authors can dock small molecules into a representative conformations of L858R and Exon 19 mutants, and into a representative conformation of a Exon 20 mutant, and hopefully show that the top hits of the former do not fare well in the latter and vice versa. This would be extremely informative to drug discovery.

3) Some of the simulation and analysis details are not described clearly or accurately, notably related to the used important collective variables. The positions of the active and inactive positions in terms of CV space seem to change across different free energy surfaces using the same CVs (See Figure 2-5). The authors need to further describe the CV they chose and explain the rationales behind the choices.

4) Three collective variables (CVs) were defined in the Methods description: CV1 was the difference between two salt-bridge distances, and CV2 and CV3 were distances in the contact map space with respect to the inactive and active conformations, respectively. Were all three CVs used in running the metadynamics simulations? In the figures and related simulation analysis, only CV1 and CV2 were used and appeared to be wrongly labeled. CV2 and CV3 were defined only related to the A-loop in the Sutto and Gervasio 2013, but described differently in this study. They need to be clarified.

5) It would help to plot CVs versus time to examine the simulation convergence and replica exchange rates. They also need to be compared with the unbiased simulation results in Figure 1-S2, which may support the authors statement that "these simulations hinted at slow motions that could not be sampled even by long MD simulations". This can be included in the SI.

6) The free energy surfaces appear noisy. Why were certain energy minima ignored for more detailed characterization, e.g., ~(0.22, 1.8) and ~(0.5, 2.3) in Figure 5 and a number of others with ~3-4 kcal/mol free energy values?

7) In Figures 3 and 4 it is confusing to show two different conformations for one single energy minimum. If they are indeed different low energy states, additional CVs may need to be defined with the corresponding new free energy profiles calculated to characterize these conformations. This need to be discussed and clarified.

8) In Figure 3 for the L858R mutant, why a clear energy minimum was identified in the "active" state in the previous study, but not in the current study.

*Reviewer #1:*

The authors present a metadynamics study comparing the free energy landscape of four distinct somatic mutations in EGFR and highlight the unique ways by which oncogenic mutations alter EGFR free energy landscape. The conformational landscape of the wild type receptor and EGFR mutants have been extensively studied in previous studies, and while the identification of intermediate states for some of the mutants is interesting, much of what the authors propose mostly validates or reinforces the models already put forward in previous studies.

*Reviewer #2:*

This study has presented unbiased microsecond molecular dynamics (MD) simulations and advanced parallel tempering metadynamics (PTmetaD) simulations to examine conformational free energy landscapes of the wild-type (WT) and four oncogenic mutants of the EGFR kinase domain. Notably, the authors have applied the same techniques to simulate the same WT and L858R mutant of EGFR in a previous study (Sutto and Gervasio, 2013, PNAS). The main difference in the present study is about simulations of three new mutants: D770-N771insNPG, A763-Y764insFQEA and dΔELREA. However, all the studied mutations generally bias the protein towards the active state, i.e., activating mutations. A number of previous publications as cited by the authors have also provided detailed mechanisms into inactivation of the WT EGFR kinase (Shan et al., 2013) and conformational changes in the activating mutant enzyme (Shan et al., 2012). In this context, the present study even with extensive simulations mostly confirms previous work and the new mechanistic insights to advance our understanding of the protein function are limited.

Strengths:

1. This paper has applied extensive unbiased and advanced enhanced sampling simulations to study the EGFR kinase domain, which is critically important for developing cancer treatments.

2. Free energy profiles can be calculated from particularly the enhanced sampling simulations to characterize conformational dynamics of the WT and mutant EGFR quantitatively.

3. Distinct conformations have been identified from the free energy profiles for the WT and four mutants of EGFR. It's clear that the mutations will bias the protein towards the activation state, which can help explaining the related previous experimental findings.

Weaknesses:

1. The work is not novel in the sense that the same techniques had been applied on the same WT and L858R mutant EGFR, except that another three different mutations are investigated here.

2. Some of the simulation and analysis details are not described clearly or accurately, notably related to the used important collective variables.

3. Apparent discrepancies are found between this study and previous work on the WT and L858R mutant EGFR.

*Reviewer #3:*

This work expands the group's previous studies that used conformational free-energy calculations to survey the conformational space of EGFR kinase WT and L858R. They apply a more recent force field and show that the observations on L858R are consistent with the previous results, which helps build confidence for this approach. They extend the work to cover Exon 20 insertion EGFR in this study. The quantitative characterization of the conformational landscapes in this work is important. The landscapes of various EGFR mutants together present a rich picture of how seemingly minor mutations can alter the conformational dynamics of EGFR kinase significantly, which is likely the case for many other proteins.

This work is particularly interesting with respect to drug discovery. An often discussed vision of drug discovery is to tailor drug molecules to selectively bind to the target in its minority but unique conformation to achieve specificity and reduce toxicity. This remains conceptual because the minority conformations are typically difficult to capture by crystal structures. MD simulations often capture such conformations but reliable quantification in terms of free energy is needed to choose a conformation to pursue drug discovery and such quantification remains untrusted. This work stands out in presenting a coherent and detailed quantitative survey. One of the more important results of this work is the characterization of exon 20 EGFR, showing that the insertions "open" the kinase and alter the shape of the active site. This explains why developing exon 20 drug remains unaccomplished and suggests a strategy for drug discovery.

---

## [Author Response]

Essential Revisions:1) The authors should compare the kinetics of inactive-active transition for all four mutants and map the lowest energy path from the inactive to active state. If the change in free energy of transition in these mutants correlate to their known fold-change in catalytic activity, this would provide additional support for their model.

Following the reviewers’ suggestion, we traced the minimum energy path (MEP) that connects the Src-like inactive to active conformation according to the calculated FES for each system studied. The MEP of each system is now given in the SI (Figure 7—figure supplement 3). With the caveat that the projection of a multidimensional FE on one dimension can underestimate the height of the energy barriers, the MEPs confirm that all mutants decrease the energy barrier to sample extended-A-loop active-like conformations.

Still, it is our understanding that, once the kinase is in an active-like state, a number of other factors affect the catalytic activity, complicating the comparison of these profiles with available experiments (Yasuda et al., 2014, Gilmer et al., 2008).

Apart from the dimerisation, which is affected by the order-disorder transitions of the aC helix, the binding of the substrate and the phosphorylation of the A-loop and the C-terminal segment will also play a role (Kim et al. Biochemistry, 2012, Kovacs et al., Annu. Rev. Biochem., 2015). For instance, the phosphorylation of the A-loop, albeit not strictly necessary to propagate the signal downstream (Zhang et al., Cell, 2006), affects the flexibility of the loop (Shan et al. Cell., 2012), further stabilising the active state and affecting the catalytic efficiency both directly and indirectly.

We reasoned that a collective motion that might be more directly linked to the steady state catalytic activity might be the hinge motion. It is linked to the phosphorylation of the substrate (as shown by NMR experiments in Src, Abl and other kinases), the active to inactive transition (Shan et al., PNAS, 2012), and also the release of the products.

To check whether the mutations have an effect on the hinge motion, we plotted the time series of this distance in the (1ms long) unbiased simulations of the monomer and the new (4ms long) simulations of homodimeric EGFR. We also recalculated the free energy surfaces as a function of the distance between V786 and T903, which has been used in the past as a probe of the kinase’s degree of extension (Shan et al., PNAS, 2012).

The unbiased simulations of the different mutants in the monomeric and homodimeric form show that, the monomer explores more ‘open’ hinge conformations with respect to the dimer (receiver kinase), which is in agreement with the activating role of dimerization. Interestingly, two of the mutants (ΔΔELREA and L858R) explore more ‘closed’ hinge conformations in the dimer see Figure *7—*figure supplement 4.

The same trend is clearly visible in the FE plots where the position of the global minimum follows the same order as the catalytic efficiency, with ΔΔELREA-EGFR exhibiting the highest catalytic activity among them (29-fold higher than that for wild-type EGFR) (Gilmer et al., Cancer Res., 2008), followed by the L858R (23-fold) (Gilmer et al., Cancer Res., 2008), theA763-Y764insFQEA (9-fold) (*Yasuda et al., Sci. Trans. Med., 2014*), and lastly the D770-N771insNPG (5-fold) (Yasuda et al., Sci. Trans. Med., 2014). Given the link between the hinge motion and the catalytic activity, we find the agreement observed both in the monomers and in the heterodimers supportive of our model. Although we can only speculate about this, the compressed kinase seen in ΔΔELREA and L858R might be linked to slow release of the substrate/ligands.

2) The observation that the Exon 20 mutants furnish a more open active site is perhaps the most direct and interesting result of this study. This should be further explored. For example, the authors can dock small molecules into a representative conformations of L858R and Exon 19 mutants, and into a representative conformation of a Exon 20 mutant, and hopefully show that the top hits of the former do not fare well in the latter and vice versa. This would be extremely informative to drug discovery.

We would like to thank the reviewers for their suggestion, and we agree that the observation of possible ‘mutant specific conformations’ associated with these Ex20Ins mutations compared to the wild-type, L858R, and ΔΔELREA conformational landscapes opens the avenue for Ex20Ins selective inhibitors design. Clearly, with a number of the emerging disclosures in this regard, there have been already extensive efforts to find Ex20Ins selective compounds, and we believe the conformational landscape presented here may assist the design of novel drugs.

Nevertheless, we believe that docking is not the most suitable way to demonstrate this potential as the docking results will depend heavily on the choice of the Ex20Ins, wild-type, L858R, and ΔΔELREA conformer(s) that will be used for the docking calculation. Secondly, it has been shown that compounds, which have the same binding mode in the mutant and wild-type forms, exhibit different affinity levels for the mutant form mainly due to the change of the mutant's affinity against ATP (Yun et al., Cancer Cell, 2008, Carey et al., Cancer Res, 2006). Taking the reviewers' feedback on board, our proposal would be to highlight the importance and impact of selecting the right conformation of EGFR for new drug discovery efforts by adding to the manuscript Figure 7–figure supplement 5.

Here, the ensemble of conformations from the PTmetaD simulations of the activating mutant L858R in the ααC-in conformation is superimposed with the crystal structures of the WT-EGFR (PDB ID: 2ITY) bound with gefitinib and lapatinib (PDB ID: 1XKK), respectively. The figure shows the improved fitting of gefitinib with respect to lapatinib and the clashes of lapatinib with the mutant protein. In the bottom panel of this figure, we show an ensemble of ααC-out conformations from the PTmetaD simulations of the Ex20Ins mutant, A763-Y764insFQEA, overlayed with the same two ligands. In this case, we see that lapatinib can be accommodated better to the ααC-out pocket, whereas the fit of gefitinib is not as tight.

We hope that this figure shows the importance of targeting the right conformation of a protein when trying to design mutant specific drugs, and we believe that we will be able to use the Exon 20 conformations in the future to try and identify specific inhibitors for these mutants by applying more sophisticated simulations, like funnel metadynamics, that take into account the flexibility of the protein while a ligand binds. However, such a comprehensive study of the selectivity of different ligands against different mutant variants is beyond the scope of this current work. We have, however, included in the manuscript the following paragraph to explain how this information could be exploited for this goal:

“Through PTmetaD simulations, we explored the differences between the conformational landscapes of EGFR Ex20Ins and the WT, L858R, and ΔΔELREA mutants. Such differences are not evident from the available crystal structures where the residues of the ATP pocket appear to be identical among the different structures. We believe that the conformational ensembles from the PTmetaD simulations of the EGFR Ex20Ins with the more extended conformations that lead to a more open active site can be used to identify novel inhibitors with increased selectivity against the mutant EGFR. We should note that in such drug design efforts, using the appropriate conformation is of great importance. For example, we can already see that targeting the L858R-EGFR with ααC-out inhibitors, like lapatinib, would lead to steric clashes that are not present in the pocket of an Ex20Ins EGFR like A763-Y764insFQEA, while inhibitors like gefitinib can be accommodated better in the tighter pocket of L858R (Figure 7—figure supplement 5).”

3) Some of the simulation and analysis details are not described clearly or accurately, notably related to the used important collective variables. The positions of the active and inactive positions in terms of CV space seem to change across different free energy surfaces using the same CVs (See Figure 2-5). The authors need to further describe the CV they chose and explain the rationales behind the choices.

In the revised manuscript, we have expanded the “Metadynamics simulations details – monomeric EGFR” section to include more details on the CVs as well as the rationale behind their choice.

This specific combination was first used in (Sutto and Gervasio., PNAS, 2013) and subsequently used to explore the conformational energy landscape of other kinases. It emerged after extensive trials, whereby different sets of CVs in combination with parallel tempering were used to explore the inactive-to-active dynamics and vice-versa. The selected combination of contact maps and salt-bridge switches was the most successful in reversibly exploring the opening and closing of the A-loop and the concerted movement of the aC helix and other important elements leading (together with PT) to a converged free energy landscape reconstruction. Subsequently, we successfully used a similar combination to explore the activation of B-Raf (Marino et al., JACS, 2015) and p38a (Kuzmanic et al., *eLife*, 2017). In our extensive experience, other choices are also effective (such as RMSD-based path collective variables). However, RMSD-based CVs are computationally expensive and severely affect the performance of the molecular dynamics code.

In the present case we also performed a new set of exploratory steered-MD simulations with the new force field (now shown in Figure 7) and the contact map-based CVs used by Sutto and Gevasio, which confirmed that the chosen set could drive the active conformation to Src-like inactive one. When a set of CVs is able to explore all the relevant conformations in a steered MD it usually leads to a converged FES in combination with the expensive parallel-tempering metadynamics approach.

A contact map was applied to monitor the distances between pairs of key representative atoms, which together are supposed to drive the transition from the active to the Src-like inactive conformation and vice-versa. As mentioned in the Methods section, here, we adapted the same pairs defined by Sutto and Gervasio, in which the most consistent contacts between equilibrated WT-active and Src-like inactive structures were selected and included, among others, salt-bridge contacts involving residues of the A-loop.

The position of the active and Src-like inactive conformations (yellow and black crosses, respectively) change indeed across the different FES plots as in these plots, we map the active and Src-like inactive conformations of each mutant on the contact map space of the WT. The change in the position of the crosses in the plots results from the way the contact maps are defined and calculated. In particular, for the definition of the contact map space a reference structure is used (termed R_inactive_ and R_active_ in the CV2 and CV3 definitions) and the distance between the atoms that are involved in each contact is included in the contact map definition. Here, the reference structures are the WT-active and WT-Src-like inactive conformations. To map the position of the equivalent mutant structures on the contact map space of the WT, the distance between the atoms of each contact in the energy minimised structures of each mutant was compared to WT-reference structure and the degree of formation of each contact in the structure R was calculated using the sigmoid function that is described in the Methods section. Deviation of the position of the atoms that define each contact in each mutant from the position of the equivalent atoms in the WT structure that was used as the reference results in different positions of the crosses in the contact map space of the WT. In addition, for the FES plots, the calculated contact-map distances were normalised and, therefore, mutants that explored larger portions of the contact-map space comparatively push the position of crosses further towards zero.

4) Three collective variables (CVs) were defined in the Methods description: CV1 was the difference between two salt-bridge distances, and CV2 and CV3 were distances in the contact map space with respect to the inactive and active conformations, respectively. Were all three CVs used in running the metadynamics simulations? In the figures and related simulation analysis, only CV1 and CV2 were used and appeared to be wrongly labeled. CV2 and CV3 were defined only related to the A-loop in the Sutto and Gervasio 2013, but described differently in this study. They need to be clarified.

The labels of the CVs in the figures of the main text have been corrected and are now in accordance with their definition in the “Material and Methods” section. As we discuss in the reviewers’ comment 3, all three CVs were biased during the PTmetaD run (now clarified in the “Material and Methods” section).

Sutto and Gervasio defined a “set of 293 regular contacts to which they added 11 salt-bridge contacts involving residues of the A-loop and formed either in the active or in the inactive conformations weighting them 3 times more than a regular contact”. As we describe in the “Material and Methods” section, here, we used the exact same set of contacts as the one Sutto and Gervasio did and adapted the atom numbers for the mutants.

5) It would help to plot CVs versus time to examine the simulation convergence and replica exchange rates. They also need to be compared with the unbiased simulation results in Figure 1-S2, which may support the authors statement that "these simulations hinted at slow motions that could not be sampled even by long MD simulations". This can be included in the SI.

To assess the convergence of a metadynamics simulation, one can calculate the estimate of the free energy as a function of the simulation time. At convergence, the reconstructed profiles should be similar to one another and to a “time-independent” reconstruction. Moreover, as the PTmetaD simulation progresses and the added bias grows, the system should be able to escape the local minimum that is trapped to at the beginning and explore a new region of the phase space in all of the replicas. Following the progression of the predefined CVs over time, one can assess whether the system gets stuck to any specific region of the CV phase space. In *Figure 7—figure supplement 1* we report the time series of the three CVs that were biased during the PTmetaD simulation and the estimate of the free energy as a function of time during the simulation for one of the mutants (ΔΔELREA), but similar behaviour was observed for the rest of the studied systems.

During the unbiased simulations, the systems explored a very limited portion of the contact map space (CV2, CV3). Even the simulations that were initiated from the active conformation in which the kinase is more flexible and samples a bigger portion of the contact map and distance space (CV1) (see Figure 7—figure supplement 2), do not lead to a transition from the active to the Src-like inactive. This highlights the necessity to use enhanced sampling techniques when the computational cost of very long unbiased simulations, like the ones reported by Shan et al. (*PNAS, 2013*), are not feasible.

Regarding the replica exchange rate, given the high temperature gap we used to aid the convergence of the simulations in a reasonable computational cost, we run the PTmetaD simulations in the well-tempered ensemble. As we describe in the Methods section, in a preliminary run, we biased the potential energy so that to increase its fluctuation and reach a target exchange rate between neighbouring replicas of 15%. The deposited Gaussians from this run were loaded in the PTmetaD to maintain this exchange rate between the replicas (clarified now in the Methods section).

6) The free energy surfaces appear noisy. Why were certain energy minima ignored for more detailed characterization, e.g., ~(0.22, 1.8) and ~(0.5, 2.3) in Figure 5 and a number of others with ~3-4 kcal/mol free energy values?

Given the high number of degrees of freedom of proteins, during a metaD run, a protein samples many conformations leading to free energy surfaces with more basins than what one sees in simpler systems. Given the fact that contact maps are usually CVs that are more adequate to distinguish conformations separated by relatively large conformational changes (e.g. active vs inactive transitions), basins that are close to each other in the contact map space are expected to correspond to ensemble of conformations that are structurally similar to one another. An example of this is the ensemble of conformations that corresponds to basins α1 and α3 (Figure 2), where, as discussed in the text, the conformations in the two basins differ mainly in the interlobal distance. Exhaustive characterisation of all the visited states is beyond the scope of this work, as here we tried to find general trends and characterise dominant metastable states.

It is true though that under physiological conditions EGFR could access conformations separated by 3-4 kcal/mol energy barriers and, therefore, we have now included some discussion on the ensemble of conformations that corresponds to basin γ4 in the FES of A763-Y764insFQEA, which also shows up to be important through the MEP analysis, as well as of basin β4 in the FES of ΔΔELREA, from which again the MEP passes through. Lastly, we have now characterised the secondary basins in the FES of L858R (basin δ2).

7) In Figures 3 and 4 it is confusing to show two different conformations for one single energy minimum. If they are indeed different low energy states, additional CVs may need to be defined with the corresponding new free energy profiles calculated to characterize these conformations. This need to be discussed and clarified.

Although the CVs that we used to bias the transition from the active to the Src-like inactive and vice-versa are able to distinguish states separated by large conformational changes, in many cases they do allow us to readily distinguish subtly different conformations within the same macro-ensembles due to their degenerative nature. Therefore, we had to undertake a series of local analyses of the conformations within the basins using different features to cluster the ensemble of conformations that fall in each basin, which are more sensitive to subtle conformational changes. In particular, as we describe in the Methods section, we used the RMSD on the Cα atoms of the A-loop and the ααC-helix residues, similar to what Pan et al. (JCTC, 2014) have used, to cluster the conformations within each basin.

In the case of the D770-N771insNPG, however, projection of the FE in the (CV1, CV3) space separates the two dominant conformations that fall together in the same basin in the (CV1, CV2) projection of the FE. As these conformations fall in the same basin in the (CV1, CV2) space and, therefore, are iso-energetic in that space, in the (CV1, CV3) space, they are also iso-energetic despite the separation. The reason behind the separation is probably the big difference in the helical content of the ααC-helix in the two conformation, which is captured better by CV3.

8) In Figure 3 for the L858R mutant, why a clear energy minimum was identified in the "active" state in the previous study, but not in the current study.

In our previous study, the deepest minimum of L858R corresponded to an ensemble of conformations which are in between the active and inactive conformation (semi-closed), while L858R was also found to populate active like conformations albeit there is a thermodynamic penalty (2–4 kcal/mol) with respect to the semi-closed state.

The use of a different force field in this study compared to the one used by Sutto and Gervasio (a99SB-disp compared to Amber99SB*-ILDN), has led to the inversion of the population of the conformations in the two basins identified in the previous study. Here, active-like conformations are the most stable while the semi-closed ones are 2-3 kcal/mol higher in energy. Force field dependent stabilisation of different conformations is a known problem of MD simulations but as can be seen from the superposition of the sampled states in Author response image 1, the results are qualitatively very close.

**Author response image 1. sa2fig1:** 

Reviewer #2:In this context, the present study even with extensive simulations mostly confirms previous work and the new mechanistic insights to advance our understanding of the protein function are limited.

We understand the concern of reviewer #2 about the novelty of our study. Still, the new biomedically relevant mutants that we studied here considerably expand our understanding of the repertoire of different effects that mutants can leverage to disrupt the regulation of kinases. While in previous papers we and others mostly focussed on the effect of single point mutations on open-to-close A-loop transition and indirectly we addressed the question of dimer stability, here we address more directly the question of dimerisation and the effect of chaperones on a set of mutants that, also due to their nature, have been reported to affect these regulatory mechanisms.

To this end, in the revised manuscript we report on new simulations of the asymmetric homodimers of the WT and mutant forms. To our knowledge, such simulations haven’t been reported before for the ΔΔELREA, D770-N771insNPG, and A763-Y764insFQEA mutants, probably due to the lack of structural data regarding their corresponding dimers. The results of these simulations further support our findings coming from the monomeric simulations according to which all mutations but the D770-N771insNPG are expected to stabilise the dimeric interface of the mutants and enhance the mutants dimerisation propensity.

Finally, the use of a more recent force-field, which has been shown to be significantly more accurate in describing partially unfolded conformations is also an important aspect of this study. The fact that the free energy landscape of the WT kinase and of the L858R mutant are in agreement with the ones previously calculated with a different force fields is indeed reassuring, but by no means it was a given. Moreover, the more balanced nature of the force field used should describe the ensemble of states explored by the A-loop more accurately as well as better capture the order-to-disorder transitions of the αC helix. This is particular important in the case of the insertion and deletion mutations.